# A seamless global daily 5 km soil moisture product from 1982 to 2021 using AVHRR satellite data and an attention-based deep learning model

Yufang Zhang[1], Shunlin Liang[2], Han Ma[2], Tao He[3], Feng Tian[3], Guodong Zhang[4], Jianglei Xu[2]

[1]School of Software, Northwestern Polytechnical University, Xi'an, 710072, China
[2]Department of Geography, University of Hong Kong, Hong Kong, 999077, China
[3]Hubei Key Laboratory of Quantitative Remote Sensing of Land and Atmosphere, School of Remote Sensing and Information Engineering, Wuhan University, Wuhan, 430079, China
[4]Faculty of Geosciences and Environmental Engineering, Southwest Jiaotong University, Chengdu, 610031, China

*Correspondence to*: Shunlin Liang (shunlin@hku.hk)

**Abstract.** Soil moisture (SM) data records longer than 30 years are critical for climate change research and various applications. However, only a few such long-term global SM datasets exist, and they often suffer from large biases, low spatial resolution, or spatiotemporal incompleteness. Here, we generated a consistent and seamless global SM product from 1982 to 2021 using deep learning (DL) by integrating four decades of Advanced Very High Resolution Radiometer (AVHRR) albedo and land surface temperature products with multi-source datasets. Considering the temporal autocorrelation of SM, we explored two types of DL models that are adept at processing sequential data, including three long short-term memory (LSTM)-based models, i.e., the basic LSTM, Bidirectional LSTM (Bi-LSTM), and Attention-based LSTM (AtLSTM), as well as a Transformer model. We also compared the performance of the DL models with the tree-based eXtreme Gradient Boosting (XGBoost) model, known for its high efficiency and accuracy. Our results show that all four DL models outperformed the benchmark XGBoost model, particularly at high SM levels (> 0.4 $m^3$ $m^{-3}$). The AtLSTM model achieved the highest accuracy on the test set, with a coefficient of determination ($R^2$) of 0.987 and root mean square error (RMSE) of 0.011 $m^3$ $m^{-3}$. These results suggest that utilizing temporal information as well as adding an attention module can effectively enhance the estimation accuracy of SM. Subsequent analysis of attention weights revealed that the AtLSTM model could automatically learn the necessary temporal information from adjacent positions in the sequence, which is critical for accurate SM estimation. The best-performing AtLSTM model was then adopted to produce a four-decade seamless global SM dataset at 5 km spatial resolution, denoted as the GLASS-AVHRR SM product. Validation of the GLASS-AVHRR SM product using 45 independent International Soil Moisture Network (ISMN) stations prior to 2000 yielded a median correlation coefficient (R) of 0.73 and unbiased RMSE (ubRMSE) of 0.041 $m^3$ $m^{-3}$. When validated against SM datasets from three post-2000 field-scale COsmic-ray Soil Moisture Observing System (COSMOS) networks, the median R values ranged from 0.63 to 0.79, and the median ubRMSE values ranged from 0.044 to 0.065 $m^3$ $m^{-3}$. Further validation across 22 upscaled 9 km Soil Moisture Active Passive (SMAP) core validation sites indicated that it could well capture the temporal variations in measured SM and remained unaffected by the large wet biases present in the input

European reanalysis (ERA5-Land) SM product. Moreover, characterized by complete spatial coverage and low biases, this four-decade, 5 km GLASS-AVHRR SM product exhibited high spatial and temporal consistency with the 1 km GLASS-

MODIS SM product, and contained much richer spatial details than both the long-term ERA5-Land SM product (0.1°) and European Space Agency Climate Change Initiative combined SM product (0.25°). The annual average GLASS-AVHRR SM dataset from 1982 to 2021 is available at https://doi.org/10.5281/zenodo.14198201 (Zhang et al., 2024), and the complete product can be freely downloaded from https://glass.hku.hk/casual/GLASS_AVHRR_SM/.

# 1 Introduction

Soil moisture (SM) is an essential climate-sensitive variable that exhibits high spatial and temporal variability. It can be measured directly by in situ sensors or indirectly through model simulations or remote sensing techniques (Liang and Wang, 2020). Accurate knowledge of the spatial and temporal distribution of SM can benefit applications across various Earth system domains, including climate, hydrology, and agriculture (Dorigo et al., 2017; Peng et al., 2021). While local- to regional-scale hydrological and agricultural applications like watershed runoff modeling, evapotranspiration estimation, and

crop yield prediction demand SM products with high spatial resolution (≤ 1 km) (Hssaine et al., 2018; Schoener and Stone, 2019; Zhuo et al., 2019), continental- to global-scale climate change-related applications, such as SM trend analyses and drought monitoring, generally require long-term data availability (> 30 years), in addition to moderate spatial resolution and high accuracy (Cheng et al., 2015; Grillakis, 2019).

Long-term point-scale SM can be measured directly by in situ sensors, thus great efforts have been devoted worldwide to

deploying and maintaining a series of operational SM networks. In situ SM datasets from some networks were shared by data organizations, which were then processed and released in a harmonized format to the public by the International Soil Moisture Network (ISMN) data repository (Dorigo et al., 2021). Still, these networks are too sparse, unevenly distributed in space, and each covers a different observation period, hindering their use in large-scale applications. Currently, large-scale SM products are typically obtained through model simulations or remote sensing techniques. Driven by long-term forcing

variables, land surface models or data assimilation systems can simulate decades of spatiotemporally continuous SM products at the global scale, with an increasingly finer spatial resolution. Several commonly used SM products include those generated by the Modern-Era Retrospective analysis for Research and Applications, version 2 (MERRA-2) at 0.5° from 1980 to the present (Gelaro et al., 2017), the Global Land Data Assimilation System version 2 (GLDAS-2) at 1°/0.25° from 1948 to the present (Rodell et al., 2004), and the land component of the fifth generation of European ReAnalysis (ERA5-Land) at

0.1° from 1950 to the present (Muñoz-Sabater et al., 2021). Yet, these SM products may suffer from large uncertainties that originated from defective forcing data or imperfect model parameterization (Ling et al., 2021).

Alternatively, microwave remote sensing techniques have been utilized for SM retrieval since the 1970s (Schmugge et al., 1974). Various global SM products have been developed from a range of active or passive microwave sensors, such as the advanced scatterometer aboard the Meteorological Operational Satellites (Bartalis et al., 2007), the microwave radiation



imager on Fengyun-3 satellites (Kang et al., 2021), and the L-band radiometers on the Soil Moisture and Ocean Salinity (SMOS) and Soil Moisture Active Passive (SMAP) satellites (Chan et al., 2018; Entekhabi et al., 2010; Kerr et al., 2012; Wigneron et al., 2021). However, the temporal coverage of these single-sensor SM products is typically short, as constrained by the operational lifespan of the satellites. In this context, the European Space Agency (ESA) Climate Change Initiative (CCI) program released a long-term global SM product spanning the period since 1978, which merged multiple active and

passive microwave SM products retrieved from different satellite instruments (Dorigo et al., 2017). Despite being the longest satellite SM dataset currently available, the ESA CCI combined SM product has a relatively low spatial resolution (0.25°) and incomplete spatial coverage, which may restrict its usage in certain applications. According to Zheng et al. (2023), the percentage of missing data in the ESA CCI combined SM product ranges from 21.8% to 94.41% at the daily scale during the period from 2000 to 2020.

In contrast, optical and thermal remote sensing techniques are characterized by long observation period, rich spectral bands, and high spatial resolution, but their relatively low sensitivity to SM poses challenges in deriving long-term global SM product solely from optical and thermal satellite observations. Over the past few decades, optical and thermal datasets have been extensively employed to downscale the coarse-scale microwave or model-simulated SM products. Most of these downscaling studies empirically or physically relate vegetation and temperature parameters to SM conditions based on the

universal triangle concept (Gillies and Carlson, 1995; Merlin et al., 2012; Piles et al., 2011). A detailed review of the strengths and limitations of various SM downscaling algorithms can refer to Sabaghy et al. (2018). In recent years, machine learning models have gradually gained popularity in the downscaling of coarse-scale SM products, such as the SMAP, ERA5-Land, and ESA CCI SM products (Karthikeyan and Mishra, 2021; Zhang et al., 2023; Zheng et al., 2023), due to their flexibility to integrate multi-source datasets and ability to implicitly learn the non-linear relationships between SM and its

influencing factors. However, the above-mentioned downscaling studies primarily concentrated on enhancing the spatial resolution of SM products, typically through integrating the fine-scale Moderate Resolution Imaging Spectroradiometer (MODIS) datasets, and there is still a lack of focus on developing long-term SM products or utilizing the four-decades Advanced Very High Resolution Radiometer (AVHRR) observations for long-term SM estimation.

Compared with conventional machine learning models, deep learning (DL) models can automatically extract relevant

features from raw datasets and learn complex non-linear relationships between variables, without the need for careful feature engineering (LeCun et al., 2015). Recently, significant progress has been made in applying DL techniques to a range of environmental remote sensing research areas, including land cover mapping, data fusion and downscaling, and environmental parameter retrieval (Yuan et al., 2020). In terms of SM retrieval, Fang et al. (2017) first utilized a long short-term memory (LSTM) model to predict spatiotemporally continuous SM over the Continental U.S., with atmospheric

forcings, modeled SM, and static attributes employed as input features, and the SMAP SM product serving as the training target. Since then, various DL models have been used in SM estimating (Gao et al., 2022; Sungmin and Orth, 2021), downscaling (Xu et al., 2022; Zhao et al., 2022), forecasting (Fang and Shen, 2020; Li et al., 2022), and gap-filling researches (Zhang et al., 2022; Zhou et al., 2023). Among them, the most frequently used DL models were the LSTM-based





models designed to capture temporal information from sequential data and the convolutional neural network (CNN) based
models constructed to extract spatial patterns from grid data, alongside several other models such as the deep neural network
and deep belief network. In those studies, input features might include brightness temperature, surface reflectance,
meteorological forcings, terrain and soil properties, land cover, precipitation, and land surface temperature (LST), depending
on the types of models they aimed to simulate, such as radiative transfer models, downscaling models, or land surface
models, while the training target varied from point-scale in situ SM to coarse-scale microwave or simulated SM. Despite the
diversity of data sources, research areas, and neural networks, all of those DL models achieved satisfactory performance,
demonstrating their good fitting and generalization capabilities, as well as great potential for generating global SM products.
Validation of those DL models against the ISMN in situ SM dataset showed that the average correlation coefficient (R)
ranged from 0.672 to 0.715, and the unbiased root mean square error (ubRMSE) ranged from 0.041 to 0.061 $m^3$ $m^{-3}$ (Gao et
al., 2022; Xu et al., 2022; Zhang et al., 2022). Nevertheless, there is still a lack of research that utilizes DL models to
generate long-term global SM data records, as evident from Table 1. Besides, while Transformer has demonstrated
effectiveness in domains like runoff modeling, drought forecasting, and crop mapping (Amanambu et al., 2022; Xu et al.,
2020; Yin et al., 2022), its application in SM estimation remains scarce.

**Table 1** Main characteristics of currently available long-term (> 30 years) global SM products.

| Category | SM products | Spatial resolution | Temporal coverage | Spatial integrity | References |
|---|---|---|---|---|---|
| Microwave | ESA CCI | 0.25° | 1978–2022 | incomplete | Dorigo et al. (2017) |
| Reanalysis | GLDAS-2 | 1°/0.25° | 1948–present | Seamless | Rodell et al. (2004) |
| | MERRA-2 | 0.5° | 1980–present | Seamless | Gelaro et al. (2017) |
| | ERA5-Land | 0.1° | 1950–present | Seamless | Muñoz-Sabater et al. (2021) |
| DL-based | GLASS-AVHRR | 5 km | 1982–2021 | Seamless | This study |

In this context, we aim to develop a long-term global SM estimation framework based on deep learning using mainly the
long-archived AVHRR satellite observations. Specifically, the AVHRR albedo and LST products from the Global LAnd
Surface Satellite (GLASS) product suite, ERA5-Land reanalysis SM product, as well as auxiliary terrain and soil texture
datasets are used as inputs, and the global 1 km GLASS-MODIS SM product generated by Zhang et al. (2023) is used as the
target to train different types of DL models. In particular, three LSTM-based models, i.e., the basic LSTM, Bidirectional
LSTM (Bi-LSTM), and Attention-based LSTM (AtLSTM), along with a Transformer model, all of which are adept at
processing sequential data, are explored. Then the best-performing model is employed to generate a four-decade
spatiotemporally continuous global SM dataset at 5 km resolution, denoted as the GLASS-AVHRR SM product. The
specific objectives of this study are:

(1) To develop a DL-based global SM estimation model using a large number of evenly distributed training samples
across the globe, so as to derive a consistent and reliable long-term global SM product;



(2) To compare the performance of different DL models: the basic LSTM, Bi-LSTM, AtLSTM, and Transformer, with the benchmark XGBoost model, and to investigate the effect of input sequence length on model accuracy;

  (3) To fully evaluate the accuracy and spatiotemporal consistency of the derived long-term GLASS-AVHRR SM product through validation against in situ SM datasets at different spatial scales and intercomparison with other long-term global SM products.

## 2 Datasets

The multi-source datasets used in this study to develop the long-term SM estimation model are summarized in Table 2. The input variables were extracted from the GLASS-AVHRR albedo and LST products, the ERA5-Land reanalysis SM product, the Multi-Error-Removed Improved-Terrain (MERIT) DEM, and the SoilGrids datasets, respectively, while the target variable was obtained from the GLASS-MODIS SM product. This section also introduces the ISMN, COsmic-ray Soil

Moisture Observing System (COSMOS), and SMAP Core Validation Sites (CVSs) in situ SM datasets used for validation, alongside the long-term ESA CCI SM product used for intercomparison.

**Table 2** Summary of the multi-source datasets used to develop the long-term SM product.

| Dataset | Variable | Temporal resolution | Spatial resolution | Usage | References |
|---------|----------|---------------------|--------------------|-------|------------|
| GLASS-AVHRR | Albedo | 8 d | 5 km | input | Qu et al. (2014);Liu et al. (2013) |
|  | LST | daily | 5 km | input | Jia (2023) |
| ERA5-Land | SM | hourly | 0.1° | input | Muñoz-Sabater et al. (2021) |
| MERIT DEM | elevation, slope, aspect | - | 90 m | input | Yamazaki et al. (2017) |
| SoilGrids | clay, sand, silt | - | 250 m | input | Poggio et al. (2021) |
| GLASS-MODIS | SM | daily | 1 km | target | Zhang et al. (2023) |

### 2.1 GLASS-AVHRR albedo and LST products

As part of the GLASS product suite, the GLASS-AVHRR albedo and LST products are generated mainly from the long-

archived AVHRR satellite observations dating back to the 1980s and are characterized by long-term temporal coverage, spatial continuity, and high accuracy (Liang et al., 2021). In particular, the GLASS-AVHRR albedo product was retrieved from the AVHRR surface reflectance through a direct estimation algorithm (Qu et al., 2014) and a spatiotemporal filtering algorithm (Liu et al., 2013). This product has been fully evaluated using ground measurements and alternative albedo products (He et al., 2014). The latest version (V5) of the GLASS-AVHRR albedo product at 5 km spatial resolution can be

downloaded from http://www.glass.umd.edu/Albedo/MIX/. Here, the black-sky visible, near-infrared, and shortwave albedo were extracted and used as input variables, with the original 8-day temporal resolution interpolated to daily to align with the training target. Meanwhile, the global all-sky GLASS-AVHRR LST product was estimated using a surface energy balance-
based algorithm (Jia, 2023), which will be released soon. The daily mean LST at 5 km resolution was also used here as an input variable.

## 2.2 ERA5-Land SM product

The ERA5-Land is a long-term state-of-the-art reanalysis dataset that includes multiple variables related to water and energy cycles spanning from 1950 to the present (Muñoz-Sabater et al., 2021). By combining the interpolated ERA5 atmospheric forcing with the European Centre for Medium-Range Weather Forecasts (ECMWF) land surface model, it achieves a seamless global coverage at an hourly temporal resolution and 0.1° spatial resolution. According to previous validation studies, while both the ERA5 and ERA5-Land reanalysis SM products typically showed high temporal correlations with in situ SM datasets, they often exhibited large biases as well (Gao et al., 2022; Li et al., 2020; Zheng et al., 2022). Here, the first-layer (0–7 cm) ERA5-Land SM product was downloaded from https://cds.climate.copernicus.eu/. The daily mean SM was then calculated and up-sampled to 5 km through bilinear interpolation before being used as an input variable for the model to provide SM background information. Moreover, the ERA5-Land SM product was also validated against in situ SM datasets and intercompared with the generated GLASS-AVHRR SM product.

## 2.3 Terrain and soil texture datasets

Topography and soil properties are the main factors that affect the spatial distribution of SM at fine scales. Here, we used the MERIT DEM (http://hydro.iis.u-tokyo.ac.jp/~yamadai/MERIT_DEM/), a high accuracy DEM generated by integrating multiple spaceborne DEMs (Yamazaki et al., 2017). This dataset covers 90°N–60°S over land at a resolution of 90 m and shows significant improvement in flat regions compared to previous spaceborne DEMs, such as the Shuttle Radar Topography Mission DEM. After downloading the MERIT DEM, it was then used to derive elevation, slope, and aspect. Meanwhile, we also used the 250-m SoilGrids product (https://www.isric.org/explore/soilgrids), a high-resolution soil property dataset generated from global soil profiles and environmental variables using machine learning models (Poggio et al., 2021). Specifically, the mean sand, silt, and clay content of the top soil layer (0–5 cm) were extracted from the SoilGrids product. All of these terrain and soil texture variables were resampled to 5 km before being used as inputs to the SM estimation model.

## 2.4 GLASS-MODIS SM product

The training target used in this study was the global 1 km spatiotemporally continuous GLASS-MODIS SM product, which was generated using an XGBoost machine learning model that integrated the GLASS-MODIS albedo, LST, and leaf area index (LAI) products with multi-source datasets. In situ SM from the representative ISMN stations distributed globally was utilized by the XGBoost model as training target. A detailed description of the development and evaluation process of the GLASS-MODIS SM product can be found in Zhang et al. (2023). This product exhibits high spatial and temporal consistency with both the ESA CCI and SMAP/Sentinel-1 L2 Radiometer/Radar SM products, while maintaining a more



complete spatial coverage. The daily GLASS-MODIS SM product from 2000 to 2020 is freely available at
http://glass.umd.edu/soil_moisture/. Here, we derived training samples from the 5 km resampled GLASS-MODIS SM
product rather than directly using in situ SM as training target, as the global SM product could provide a much richer and
representative training set than the sparse ISMN SM dataset.

**2.5 In situ SM datasets**

After generating the GLASS-AVHRR SM product using the developed DL model, three types of in situ SM datasets at
different spatial scales were adopted to evaluate its accuracy and consistency. The characteristics of these in situ SM datasets
are listed in Table 3, and the spatial distribution of the corresponding SM stations is shown in Fig. A1. The first type is the
ISMN dataset, which consists of harmonized and quality-controlled in situ SM measurements collected from over 2800
monitoring sites worldwide (Dorigo et al., 2021). This point-scale SM dataset covers a period from 1952 to the present,
providing a valuable reference for validating satellite-based and model simulated SM products, despite the relatively poor
spatial representativeness of some SM stations. There were 1672 ISMN stations available for validation during Period I
(2000–2018). Among them, 715 spatially representative stations were selected using the triple collocation method, as
described in detail in Zhang et al. (2023). Although SM datasets from these representative stations were previously used as
target to train the GLASS-MODIS SM estimation model, making them only partially independent, they can be used here to
assess the consistency in accuracy between the GLASS-AVHRR and GLASS-MODIS SM products. Moreover, the 45 fully
independent ISMN stations from Period II (1982–1999) can be used to evaluate the accuracy of the GLASS-AVHRR SM
product during the earlier years. The daily mean SM was calculated by averaging the hourly SM measurements at the top
soil layer (0–5 cm) obtained from https://ismn.earth/, considering only those with a quality flag of "G".

The second type is the COsmic-ray Soil Moisture Observing System (COSMOS) SM dataset, which includes area-averaged
SM measurements at the field scale from three COSMOS networks: COSMOS (Zreda et al., 2012), COSMOS-UK (Cooper
et al., 2021), and COSMOS-Europe (Bogena et al., 2022). The COSMOS sensors detect low-energy cosmic-ray neutrons
above the ground, which can be converted to SM within a footprint radius of 130 to 240 m and a penetration depth of 15 to
83 cm, depending on factors such as air humidity, SM, and vegetation (Köhli et al., 2015). Although data from the COSMOS
and COSMOS-UK networks had been integrated into the ISMN database, they were excluded from the training dataset of
the GLASS-MODIS SM estimation model because their observation depths exceeded the 5 cm threshold. Recently, data
from the COSMOS-Europe network have been released and can be accessed at https://doi.org/10.34731/x9s3-kr48.
Collectively, these post-2000 SM datasets can serve as an independent source for validating the GLASS-AVHRR SM
product at an intermediate scale. After filtering based on the quality flags and aligning with the GLASS-AVHRR SM
product, there were 102 COSMOS, 45 COSMOS-UK, and 51 COSMOS-Europe stations available for validation.

The third type is the SMAP/In situ core validation site (CVS) match-up dataset, which contains the up-scaled in situ SM
measurements derived from multiple quality-controlled stations that have been aligned with SMAP SM products (Colliander
et al., 2017). A total of 22 globally distributed CVSs were matched with the SMAP-Sentinel L2 SM product gridded at 9 km



resolution (SMAPL2SMSP9km). This independent 9 km SMAP CVS in situ dataset can be used to validate the GLASS-AVHRR SM product with reduced impact of scale difference. It covers the period from 2015 to the present and can be downloaded from https://nsidc.org/data/nsidc-0712/versions/1.

**Table 3** Characteristics of three types of in situ SM datasets used in this study at different spatial scales.

| Dataset | Group of stations | No. of stations | Spatial scale | Sensing depth | Time period | References |
|---------|-------------------|-----------------|---------------|---------------|-------------|------------|
| ISMN | All ISMN (Period I) | 1672 | Point-scale | 0–5 cm | 2000–2018 | Dorigo et al. (2021) |
| | Representative ISMN (Period I) | 715 | | | 2000–2018 | |
| | ISMN (Period II) | 45 | | | 1982–1999 | |
| COSMOS | COSMOS | 102 | 130–240 m | 15–83 cm | 2008–2018 | Zreda et al. (2012) |
| | COSMOS-UK | 45 | | | 2013–2018 | Cooper et al. (2021) |
| | COSMOS-Europe | 51 | | | 2011–2018 | Bogena et al. (2022) |
| CVS | SMAP CVS | 22 | 9 km | 0–5 cm | 2015–2021 | Colliander et al. (2017) |

**2.6 ESA CCI SM product**

The European Space Agency (ESA) launched the Climate Change Initiative (CCI) SM project to develop the ESA CCI SM dataset, a global daily multi-decadal dataset aimed at supporting climate research (Dorigo et al., 2017). This dataset merged multiple microwave SM products into active-only, passive-only, and combined active-passive products, respectively. The

evolution of the merging algorithm and break correction method are described in detail by Gruber et al. (2019) and Preimesberger et al. (2021). Here, we used the ESA CCI SM v7.1 combined product at a resolution of 0.25° (https://climate.esa.int/en/projects/soil-moisture/data/), which covered the period 1978–2021. Despite being the most widely used long-term satellite SM product, it suffers from spatial incompleteness due to the lack of satellite observations in the earlier years, the observation gaps in satellite orbits, and the physical limitations of microwave observations for SM retrieval

over densely vegetated areas (Dorigo et al., 2017). In this study, the spatial consistency between the ESA CCI combined SM product and our GLASS-AVHRR product was investigated.

**3 Methods**

Figure 1 shows the flowchart of the proposed long-term global GLASS-AVHRR SM estimation framework, which consists of three main parts: data preprocessing and training samples preparation, model training and performance comparison, and

generation and evaluation of the GLASS-AVHRR SM product. In particular, the multi-source datasets and their pre-processing can refer to Sect. 2; the preparation of training samples, description of five data-driven models, and adopted





evaluation methods are given in Sect. 3; the detailed comparison of model performance and evaluation of the long-term SM product are presented in Sect. 4.

**Figure 1** Flowchart of the proposed long-term global GLASS-AVHRR SM estimation framework

**3.1 Training samples**

The global GLASS-MODIS SM product resampled at 5 km was used as the training target of the long-term SM estimation model, from which a large number of representative and evenly distributed training samples could be obtained. Considering that the size of training samples would be too large if all the pixels were included, these samples were selected at 25 km (5 240 pixels) intervals along both the longitude and latitude, and a total of 135,360 pixels were chosen after excluding those with a large proportion of missing values. Based on the geographic coordinates of these pixels, the values corresponding to each

input feature as well as the target SM for the years 2005, 2010, and 2015 were extracted, which collectively formed the time-series training samples. These samples were then randomly divided into training dataset, validation dataset, and test dataset in the ratio of 7:2:1 according to their locations. While the training and validation datasets were used to train and tune the

hyperparameters of the models, the accuracy of the models was evaluated on the test dataset. Note that, the input features need to be scaled before training a DL model, which helps to speed up the convergence process, avoids bias towards larger-scale features, and improves the model stability. Here, each input feature was standardized by subtracting the mean and then dividing by the standard deviation, whereas for the target SM, no further processing is needed as it is by definition scaled.



**Figure 2** Schematic diagrams of the five models used in this study: (a) LSTM, (b) Bi-LSTM, (c) AtLSTM, (d) Transformer,
and (e) XGBoost. In subplots (a-d), $x_t$, $y_t$, and $h_t$ represent the input datasets, SM prediction, and hidden state output by the models at time step $t$, respectively.

**3.2 Benchmark model**

When generating the global 1 km GLASS-MODIS SM product, an XGBoost model was employed to integrate the multi-source datasets because of its good performance and high training and predicting speed. Here, we used the XGBoost model





as a benchmark and compared its performance with the DL models (LSTM-based and Transformer) to analyze whether the DL models that account for temporal information exhibit an advantage over this widely used tree-based conventional machine learning model in SM estimation. The XGBoost model (Chen and Guestrin, 2016) is a type of gradient boosting model, in which multiple weak learners (trees) are iteratively constructed through correcting the prediction residuals of the preceding trees. The schematic diagram of the XGBoost model is shown in Fig. 2e, where predictions from multiple trees are combined to make the final SM prediction. The open-source XGBoost Python package was utilized for model training, with the key hyperparameters configured as follows: n_estimators (number of trees) = 1000, learning_rate = 0.1, and max_depth (maximum depth of tree) = 8. The time-series training samples constructed above were put together to train the XGBoost model, and the overall accuracy achieved by the XGBoost model on the test dataset was then compared with that of the DL models as a benchmark.

### 3.3 Long short-term memory-based models

The LSTM network (Hochreiter and Schmidhuber, 1997) is a special type of recurrent neural network (RNN) designed to solve the problems of gradient vanishing and exploding when training long sequences, where the LSTM network can outperform the original RNN. The basic LSTM network introduces the memory cell, which is a special type of hidden state that shares the same shape as the hidden state but is designed to record long-term information. Each recurrent unit within the LSTM has three distinct gates, i.e., the forget gate, input gate, and output gate, as illustrated in Fig. 2a. These gates work together to regulate the flow of information through the LSTM network, facilitating the update of its memory cell, thus enabling the network to selectively retain or discard information over time. The formulas used to calculate the three gates ($f_t, i_t, o_t$), cell state ($c_t$), and hidden state ($h_t$) are given below:

$$f_t = \sigma(W_f . [h_{t-1}, x_t] + b_f) \tag{1}$$

$$i_t = \sigma(W_i . [h_{t-1}, x_t] + b_i) \tag{2}$$

$$o_t = \sigma(W_o . [h_{t-1}, x_t] + b_o) \tag{3}$$

$$c_t = f_t * c_{t-1} + i_t * \tanh(W_c . [h_{t-1}, x_t] + b_c) \tag{4}$$

$$h_t = o_t * \tanh(c_t) \tag{5}$$

where $x_t$ represents the input datasets at time step $t$, $h_{t-1}$ is the hidden state at the previous time step; $f_t, i_t$, and $o_t$ are all calculated as linear functions of $x_t$ and $h_{t-1}$ with different weights and biases, and are then rescaled using a non-linear sigmoid ($\sigma$) function. The $\sigma$ function acts as the gating function for the three gates with an output ranging between 0 and 1, thereby determining which portion of the information passes through the gates. The tanh activation function is used to rescale the values to the range between -1 and 1 during the calculation of $c_t$ and $h_t$, helping to avoid the vanishing gradient problem. Both the $\sigma$ and tanh functions add non-linearity to the LSTM network. The Bidirectional LSTM (Bi-LSTM) extends the LSTM network by using two separate LSTM layers to process the input sequence from both forward and



backward directions, and then concatenating the outputs of both layers. As displayed in Fig. 2b, the Bi-LSTM model can learn bidirectional (preceding and following) information at each time step.

The LSTM network has different application architectures, including many-to-one (MTO) and many-to-many (MTM). In research areas like crop mapping, runoff prediction, and SM forecasting, the MTO architecture is primarily adopted, where inputs from multiple time steps are fed into an LSTM network, and then estimates for a single time step (typically the last time step) are output. Alternatively, we choose to use the MTM architecture, where time-series input features are fed into the network, and time-series SM estimates are output at once. This was implemented simply through feeding the hidden states output by the LSTM network at all time steps into a fully connected layer, thereby simultaneously obtaining SM estimates across all time steps. We also designed an experiment to compare the difference in accuracy between these two architectures in estimating SM.

In addition to the basic LSTM and Bi-LSTM networks introduced above, an attention module was added to the Bi-LSTM network, referred to as the AtLSTM network, to explore if the estimation accuracy of SM could be further improved. The AtLSTM network was constructed based on Bahdanau et al. (2016) and Xu et al. (2020), and adapted here for the MTM architecture. As illustrated in Fig. 2c, the attention module generates the attention weights ($\alpha$), which are then multiplied with the hidden states ($h$) to get the weighted hidden states ($h^*$). The $\alpha$ and $h^*$ can be calculated as follows:

$$e_t = W_a . h_t + b_a \tag{6}$$

$$\alpha_{t,i} = softmax(e_{t,i}) = \frac{exp\,(e_{t,i})}{\sum_{j=1}^{T} exp\,(e_{t,j})} \tag{7}$$

$$h_t^* = \sum_{i=1}^{T} \alpha_{t,i} * h_i \tag{8}$$

where $W_a$ and $b_a$ denote the learnable parameters that map the hidden states $h$ into a weight matrix $e$, and $T$ is the sequence length of the input features. The weight matrix (with the shape of $T \times T$) is then rescaled by a softmax function to obtain the attention weights for each hidden state, which range between 0 and 1 and sum to 1. The softmax function can also enhance the importance of elements with higher values in the weight matrix. The weighted hidden states $h^*$ are finally calculated as the matrix multiplication between the attention weights $\alpha$ and original hidden states $h$, which are then fed into a fully connected layer to estimate the target variable. Intuitively, the attention weights can reflect the importance of the hidden state at each time step relative to the current hidden state. Higher attention weights indicate that the corresponding hidden states have a greater influence on the estimation of SM at a specific time step.

In this study, the LSTM-based models were implemented using the open-source PyTorch 2.0 framework. The mean square error (MSE) between the models' output and target SM was used as the loss function. The Adam optimizer was adopted to update the learnable parameters of the models, such as the weights and biases in Eq. (1-6), to minimize the loss function during the training phase. Several key hyperparameters were tuned, including the hidden size, number of epochs, and learning rate (Zhang et al., 2021). For each model, the hidden size was determined after testing values of 64, 128, 256, and 512; the number of epochs after testing 20, 50, 100, and 200; and the learning rate after testing 0.1, 0.01, 0.001, and 0.0001. The final settings of the major hyperparameters for the three LSTM-based models are listed in Table 4.





**Table 4** Key hyperparameters configured for the DL models used in this study.

| Hyperparameters | LSTM | Bi-LSTM | AtLSTM | Transformer |
|---|---|---|---|---|
| Hidden size | 256 | 256 | 256 | 64 |
| Number of heads | / | / | / | 4 |
| Number of epochs | 100 | 100 | 200 | 100 |
| Number of layers | 2 | 1 | 1 | 1 |
| Batch size | 100 | 100 | 100 | 100 |
| Learning rate | 1e-3 | 1e-3 | 1e-4 | 1e-3 |
| Sequence length | 425 | 425 | 425 | 365 |

**3.4 Transformer**

The Transformer network is a DL architecture based entirely on attention mechanisms, dropping the use of recurrent structure to avoid the constraint of sequential calculation. After being proposed by Vaswani et al. (2017), Transformer has soon become the state-of-the-art neural network for natural language processing tasks, and it has also been successfully

applied to many other domains such as computer vision (Dosovitskiy et al., 2021) and time series analysis (Wen et al., 2023). The core component of the Transformer network is the multi-head self-attention layers, which can relate any two positions in a sequence to generate representations of the sequence. More specifically, multi-head attention involves applying the attention function to multiple sets of key, value, and query vectors in parallel, thus enabling the model to focus on information from different parts of the input sequence simultaneously. Unlike the attention function used in the AtLSTM

model (Eq. (6-7)), the self-attention function used by the Transformer network is called the scaled dot-product attention $\alpha$, which can be calculated as follows:

$$\alpha\left(Q, K, V\right) = softmax\left(\frac{QK^T}{\sqrt{d_k}}\right)V \tag{9}$$

where $Q$, $K$, and $V$ refer to the query, key, and value vectors, respectively, which are derived by multiplying the embedded input sequence with the corresponding learnable projection matrix; $d_k$ is the dimension of the key and query vectors.

Additionally, with the help of a positional encoding function, the Transformer network can retain some ordinal information for elements in the input sequence. A detailed description of Transformer and the multi-head self-attention mechanism can be found in Vaswani et al. (2017). Compared with recurrent or convolutional neural networks, the Transformer network can efficiently parallelize much larger amounts of computation and capture long-range dependencies in the input sequence more easily. Here, we only used the encoder portion of the original Transformer network to map the input features into hidden

representations, which were then fed into a fully connected layer to output the time-series SM estimates, as displayed in Fig. 2d. The same training samples, optimizer, and loss function used for the LSTM-based models were employed to train the Transformer network, with the settings of its hyperparameters also listed in Table 4. In particular, the number of heads is a unique hyperparameter of Transformer that refers to the number of parallel self-attention layers of the encoder.



## 3.5 Evaluation of the models and GLASS-AVHRR SM product

After training the benchmark XGBoost model and the four DL models described above using the same training samples distributed worldwide, their performance on the test set was then compared from multiple perspectives, including comparisons between the DL models and XGBoost model, between the DL models with different attention mechanisms, and between the DL models with MTM or MTO architectures. Moreover, the effect of input sequence length on model accuracy was investigated using the LSTM-based models, and a preliminary interpretability analysis was performed through

visualizing the attention weights of both the AtLSTM and Transformer models. Then, the best-performing model, along with the multi-source input datasets, was employed to generate the global daily GLASS-AVHRR SM product at 5 km resolution from 1982 to 2021. To fully assess the derived long-term SM product, different SM datasets and evaluation strategies were combined, including overall accuracy evaluation, scatter plots analysis, time-series plots comparison, and spatial consistency examination. Specifically, the accuracy of this product was first evaluated against the point-scale ISMN, field-scale

COSMOS, and upscaled 9 km SMAP CVS in situ SM datasets, respectively. Then, the GLASS-AVHRR SM product was intercompared with the GLASS-MODIS SM product and two widely used long-term global SM products to investigate their spatial consistency.

## 4 Results

### 4.1 Comparison of model performance

Table 5 lists the performance metrics achieved by the benchmark tree-based XGBoost model and four DL models on the training set, validation set, and two types of test sets, respectively. The XGBoost model achieved similar overall accuracy across the training, validation, and test sets, with a coefficient of determination ($R^2$) of 0.984 and RMSE of 0.012 m$^3$ m$^{-3}$ on the training set, and an $R^2$ of 0.982 and RMSE of 0.013 m$^3$ m$^{-3}$ on both the validation and test sets, indicating a low tendency for overfitting. The fairly high overall accuracy attained by the benchmark XGBoost model may be attributed to the large

number of training samples that are evenly distributed across the globe on a daily basis, specifically 135,360 pixels per day for 3 years. When the size of training samples was reduced by a factor of 100, the accuracy of the XGBoost model dropped considerably, with an $R^2$ of 0.96 and RMSE of 0.017 m$^3$ m$^{-3}$ on the test set. Meanwhile, Table 5 also shows that the accuracy of the XGBoost model decreases drastically on the test set with SM observations exceeding 0.4 m$^3$ m$^{-3}$, yielding an $R^2$ of 0.413 and RMSE of 0.022 m$^3$ m$^{-3}$, likely due to the relatively smaller portion of samples at high SM levels.

In comparison, the LSTM model developed using time-series training samples performed slightly better than the XGBoost model, with the $R^2$ on the test set increasing to 0.983, and when the Bi-LSTM model was employed, the overall accuracy on the test set was further improved, with the $R^2$ increasing to 0.985 and RMSE decreasing to 0.012 m$^3$ m$^{-3}$. Although the increase in the overall accuracy might not be significant, the Bi-LSTM model exhibited significant improvement over the XGBoost model at high SM levels, achieving an $R^2$ of 0.482 and RMSE of 0.020 m$^3$ m$^{-3}$ on the test set for observations



Earth System
Science
Data

exceeding 0.4 $m^3$ $m^{-3}$. As also can be seen from the density scatter plots in Fig. 3, the majority of samples had SM values below 0.4 $m^3$ $m^{-3}$ (indicated by the red dots), where all models achieved high prediction accuracy. However, on the relatively infrequent samples with high SM values, where the XGBoost model tended to underestimate, both the LSTM and Bi-LSTM models provided more accurate estimates. Given the temporal autocorrelation of SM, these results suggest that learning both forward and backward temporal information from the time-series training samples enhances the ability of DL models to

estimate SM more accurately, especially at high SM levels with sparser samples.

**Table 5** Performance metrics of the benchmark XGBoost model and four DL models on the training set, validation set, and two types of test sets, respectively.

| Model | Training set | | Validation set | | Test set | | Test set (> 0.4 $m^3$ $m^{-3}$) | |
|---|---|---|---|---|---|---|---|---|
| | $R^2$ | RMSE ($m^3$ $m^{-3}$) | $R^2$ | RMSE ($m^3$ $m^{-3}$) | $R^2$ | RMSE ($m^3$ $m^{-3}$) | $R^2$ | RMSE ($m^3$ $m^{-3}$) |
| XGBoost | 0.984 | 0.012 | 0.982 | 0.013 | 0.982 | 0.013 | 0.413 | 0.022 |
| LSTM | 0.986 | 0.012 | 0.983 | 0.013 | 0.983 | 0.013 | 0.424 | 0.021 |
| Bi-LSTM | 0.988 | 0.011 | 0.984 | 0.012 | 0.985 | 0.012 | 0.482 | 0.020 |
| AtLSTM | 0.990 | 0.010 | 0.986 | 0.011 | 0.987 | 0.011 | 0.621 | 0.016 |
| Transformer | 0.990 | 0.010 | 0.984 | 0.012 | 0.985 | 0.012 | 0.460 | 0.021 |

Then, after adding the attention module into the Bi-LSTM model, the derived AtLSTM model achieved the best performance, with an $R^2$ of 0.987 and RMSE of 0.011 $m^3$ $m^{-3}$ on the test set. In contrast, despite that the Transformer model also

incorporated an attention module, its accuracy was slightly lower than that of the AtLSTM model on the test set and significantly lower on samples with high SM levels (> 0.4 $m^3$ $m^{-3}$) in our experiments. As mentioned above, the main advantage of the Transformer model is its ability to capture long-range dependencies and handle long sequences effectively. However, soil moisture often exhibits high temporal variability, meaning it can change rapidly due to factors such as rainfall and evaporation. In this context, short-term adjacent temporal information can be critical for accurate SM estimation. The

slightly better performance of the AtLSTM model compared with the Transformer model may be attributed to its superior ability to capture these short-term adjacent dependencies, which are critical for modeling the nuances in rapidly changing SM levels. This will be further investigated through the analysis of attentional weights below. Additionally, a feature importance analysis was conducted for the best-performing AtLSTM model, as shown in Fig. A2. Specifically, the gradients of the model's output with respect to each input feature were computed on the test set, and the absolute values of these

gradients were then averaged across all samples and time steps. Input features with larger average gradients are considered to exert a more significant influence on the model's predictions. The results indicate that elevation, black-sky visible albedo, ERA5-Land reanalysis SM, and slope are the most influential features for the AtLSTM model.

While the numerical differences in overall accuracy among all these models may not seem remarkable, a more intuitive comparison can be drawn from their density scatter plots. As shown in Fig. 3, on the majority of samples, both the best-

performing AtLSTM model and benchmark XGBoost model can achieve high prediction accuracy, resulting in a relatively



small difference in their overall performance on the test set. However, there remains a small portion of samples that are more challenging to predict, on which the SM estimates from the AtLSTM model are much closer to the 1:1 line compared with the XGBoost model. Furthermore, the AtLSTM model significantly improves upon the underestimation observed in the XGBoost model at high SM levels, achieving an $R^2$ of 0.621 and RMSE of 0.016 $m^3$ $m^{-3}$ on the test set for observations

exceeding 0.4 $m^3$ $m^{-3}$. Overall, while both the XGBoost model and the four DL models can achieve high SM estimation accuracy, the AtLSTM model yields the highest accuracy among them and performs well across different SM levels with a low tendency for overfitting. This suggests that utilizing bidirectional temporal information from the input sequence as well as adding an attention module are both effective in further improving the estimation accuracy of SM.

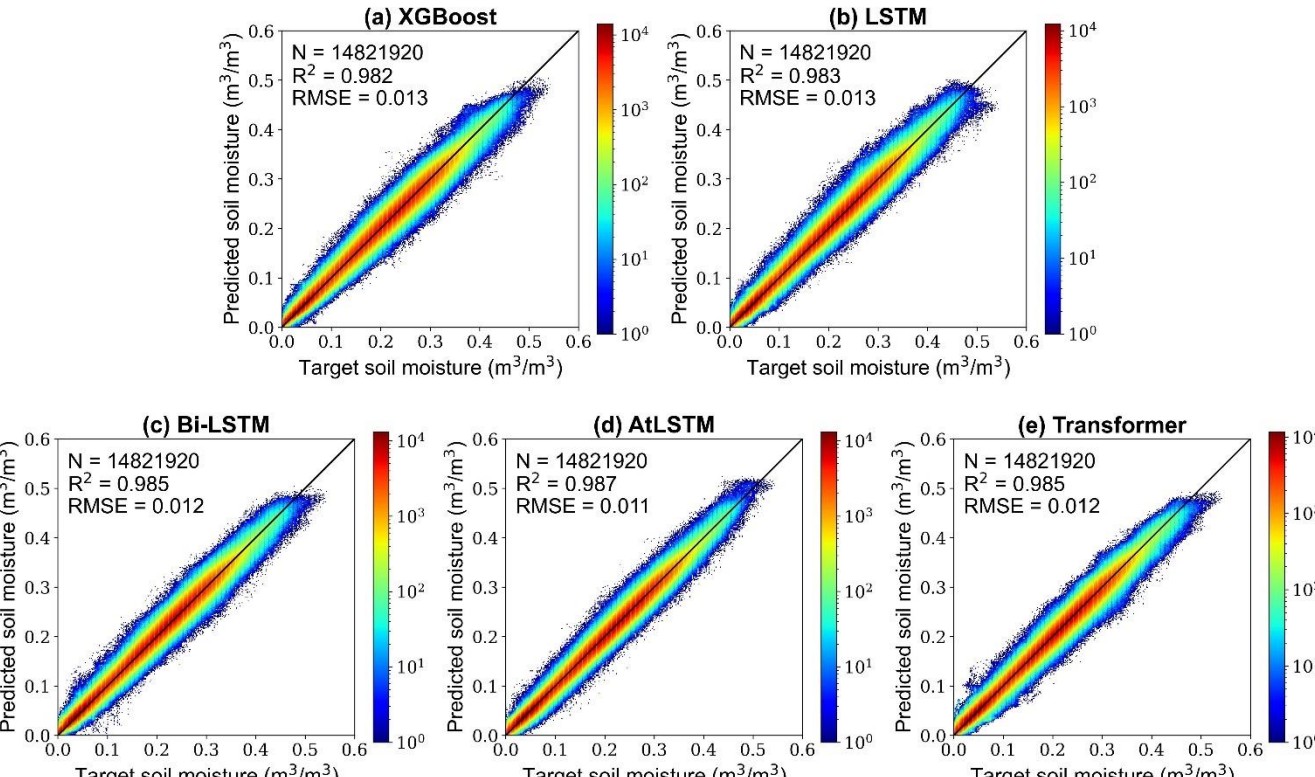

**Figure 3** Scatter plots between target SM and predicted SM for the (a) XGBoost, (b) LSTM, (c) Bi-LSTM, (d) AtLSTM and (e) Transformer models on the test set. The colors of the dots indicate different probability densities, and the black line represents the 1:1 line.

As mentioned in Sect. 3.3, We chose to use the MTM architecture when developing the DL models to output time-series SM estimates at once. Here, to compare the accuracy of the MTM architecture with the more commonly used MTO architecture,

as well as to investigate the effect of input sequence length on model accuracy, we calculated performance metrics for the LSTM models utilizing these two different architectures under varying lengths of input sequences. Specifically, both types of models were trained using input features from a given date (e.g., the first day of 2015) and n days (0-29) prior to that date,

respectively, and the accuracy of the models was then evaluated on the test set for that given date. To save training time, the number of epochs for these LSTM models was set to 20. It can be seen from the $R^2$ and RMSE curves in Fig. 4a that, as the

length of input sequence increased, the accuracy of the LSTM model with the MTO architecture also increased, and then the accuracy leveled off at a sequence length of about 10 days. This indicates that while accounting for temporal information can be beneficial for current SM estimation, only the most recent input sequences have a remarkable effect on the model's accuracy. In comparison, the LSTM model with the MTM architecture, which can output a sequence of SM estimates simultaneously, achieved similar accuracy to that of the MTO architecture, and its $R^2$ and RMSE curves stabilized at a

sequence length of about 5 days. This demonstrates the feasibility of adopting the MTM architecture in the LSTM model, which not only reduces considerable production time but also maintains the estimation accuracy.

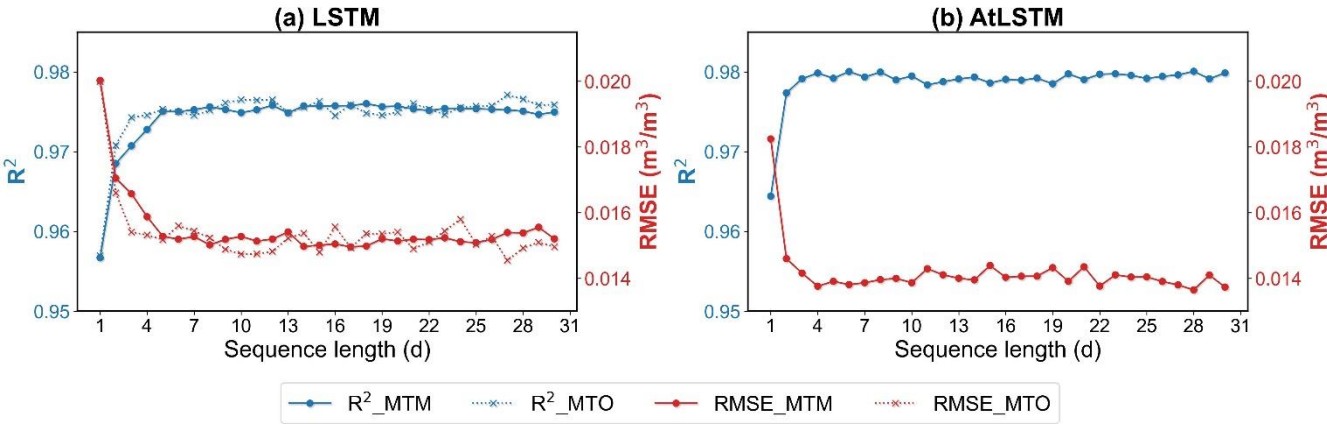

**Figure 4** Performance metrics of the (a) LSTM models with two different types of architectures (MTO and MTM) and (b) AtLSTM model with the MTM architecture trained using varying lengths of input sequences on the test set. The blue and red

curves represent the $R^2$ and RMSE curves, respectively.

Moreover, we also investigated the effect of input sequence length on the overall accuracy of the AtLSTM model with the MTM architecture, and the performance metrics were calculated here based on SM estimates over the entire time series instead of on a given date. To save training time and accounting for the smaller learning rate used for the AtLSTM model (Table 4), the number of epochs was set to 50. As displayed in Fig. 4b, the overall accuracy of the AtLSTM model increased

sharply as the length of input sequence increased, and then the accuracy plateaued at a sequence length of about 4 days. The more rapid stabilization of the AtLSTM model's accuracy may be attributed to the incorporation of the Bi-LSTM module in the model, which can utilize both forward and backward temporal information. In addition, it seems that when the input sequence is long enough, the model can automatically learn the necessary temporal information to accurately estimate SM at each position in the sequence. However, it should be noted that at the beginning or end of the sequence, the model's

accuracy tends to decrease as only forward or backward information can be utilized, which is a common issue encountered by the LSTM-based models with the MTM architecture. Therefore, to facilitate the production process, the sequence length of the LSTM-based models was finally set to 425, and both the first 30 and last 30 values were removed (a rather sufficient

discarding number) after the model output the time-series SM estimates, so that an entire year's SM estimates could be obtained in one go.

Although data-driven DL models are commonly perceived as "black boxes", there are many techniques that can be employed to increase the interpretability of DL models. In the case of attention-based deep neural networks, this can be achieved by analyzing the distribution of attention weights. In a long sequence, perhaps only a portion of the information is critical to the model prediction at a given time step, and the attention mechanism enables the model to focus on these critical positions. In particular, the attention module of the AtLSTM model can dynamically adjust the weights of the hidden states output by the

model at each time step. Figure 5a illustrates the distribution of the averaged attentions weights calculated using the best-performing AtLSTM model on the test set (40,608 samples). To show more detail, only the attention weights of 30 consecutive days selected from the entire sequence (425 days) are displayed here, and attention weights less than 0.0001 are masked out. It is observed that, for the hidden state at each time step in the sequence (vertical axis), the largest attention weight was located approximately 3 days around that time step (horizontal axis). This indicates that when the attention

module of the AtLSTM model learns to readjust the hidden states, it primarily utilizes the temporal information from adjacent positions in the sequence.

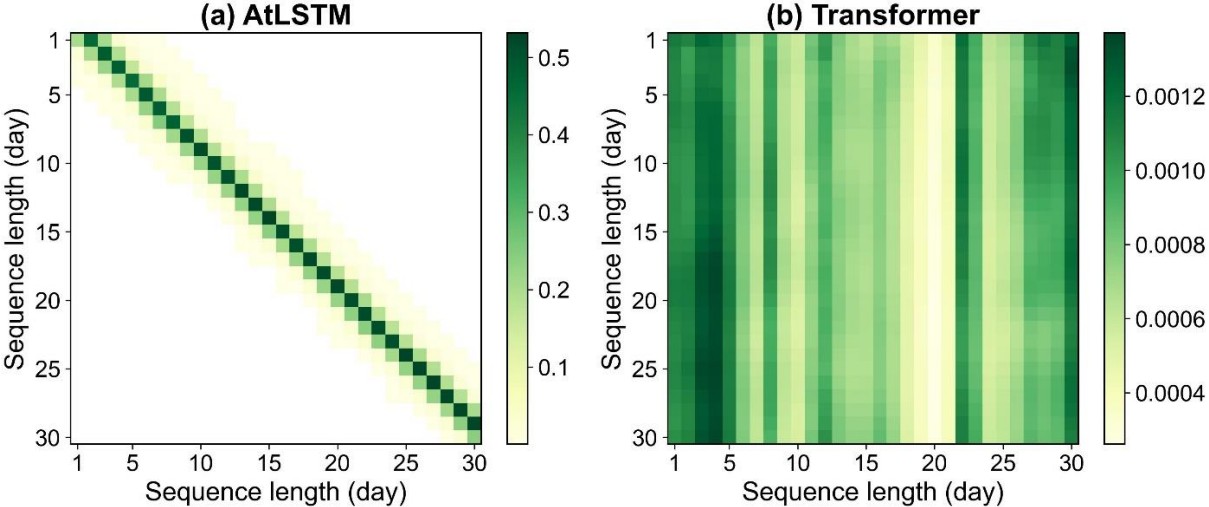

**Figure 5** Heatmaps of the averaged attention weights calculated using the (a) AtLSTM and (b) Transformer model on the test set (40,608 samples). Only the attention weights of 30 consecutive days selected from the entire sequence are displayed

here for illustration.

In contrast, as a core component of the Transformer model, the multi-head self-attention layers can capture various aspects of relationships between different positions within a sequence, and the attention weights generated by these layers are then directly applied to the embedded input sequence. Figure 5b shows, as a comparison, the distribution of attention weights calculated by averaging the outputs from the four attention heads of the Transformer model. The attention weights heatmap

of the Transformer model is quite different from that of the AtLSTM model, with the weight at each position being much



smaller and dispersed. This is likely because the self-attention module can relate any two positions in the sequence, and inputs from more distance positions may contribute more to the model output at the current time step. In addition, for each time step in the sequence (vertical axis), there were some common positions (horizontal axis) with larger weights that were more important for model prediction. Despite the distinct attention mechanisms employed by these two DL models, both of
them achieved high SM estimation accuracy. Given that SM is temporally autocorrelated and highly variable over time, the slightly better performance of the AtLSTM compared to the Transformer model may be attributed to that it extracts temporal information mainly from adjacent positions in the sequence, rather than from more distance ones, for SM estimation.

**4.2 Validation of the GLASS-AVHRR SM product**

After generating the GLASS-AVHRR SM product using the best-performing AtLSTM model with the MTM architecture,
permanent snow and ice as well as water bodies were masked out with the help of the MODIS land cover type product (MCD12C1) (Friedl and Sulla-Menashe, 2022). The derived SM product was then evaluated against three types of in situ SM datasets at different spatial scales. The first type is the point-scale ISMN SM dataset, which is distributed globally and covers a wide range of land cover types. There were 1672 ISMN stations and 715 spatially representative stations available for validation during Period I (2000–2018). The distribution of validation metrics achieved by the GLASS-AVHRR SM
product on these partially independent ISMN stations during Period I, grouped by all stations and representative stations, is presented in Fig. 6, alongside those of the GLASS-MODIS and ERA5-Land SM products for comparison. The GLASS-AVHRR SM product achieved comparable performance to that of the GLASS-MODIS SM product across all ISMN stations and representative stations during Period I. In addition, both GLASS SM products performed significantly better at the representative stations. This demonstrates the high level of consistency in accuracy between the two GLASS SM products.
Note that the validation metrics for the GLASS-MODIS product were derived using a site-independent cross-validation method, which was designed to accurately reflect the product's performance over unknown areas. Given the consistency in the distribution of validation metrics between the GLASS-AVHRR and GLASS-MODIS SM products, the accuracy achieved by the GLASS-AVHRR product at these partially independent stations should also approach its true accuracy. In contrast, although the ERA5-Land SM product achieved a similar distribution of R to the two GLASS SM products across
all ISMN stations and representative stations, it exhibited much larger biases and ubRMSE values.

To conduct a more independent evaluation of the GLASS-AVHRR SM product, the ISMN SM dataset from Period II (1982–1999) was also collected. After excluding stations that overlapped with the 715 representative stations from Period I, only 45 independent stations remained for evaluation during Period II. The observations at these stations were also quite limited, hence the validation metrics derived from them may not provide a comprehensive assessment. Nevertheless, it can be seen
from Fig. 6 that the GLASS-AVHRR product achieved rather high accuracy at these stations, with a median R of 0.73 and a median ubRMSE of 0.041 $m^3$ $m^{-3}$. Likewise, while the ERA5-Land SM product exhibited a similar distribution of R to the GLASS-AVHRR product at these stations, it achieved much larger biases and ubRMSE values. The second type of in situ SM dataset comprises field-scale measurements from three COSMOS networks: COSMOS, COSMOS-UK, and COSMOS-





Europe, which can provide an independent evaluation of the GLASS-AVHRR SM product at an intermediate scale. As
shown in Fig. 6, both the GLASS-AVHRR and ERA5-Land SM products achieved good performance across all three
COSMOS networks. Yet, their accuracies varied considerably across these networks, with the median R ranging from 0.63
to 0.79 and the median ubRMSE ranging from 0.044 to 0.065 $m^3$ $m^{-3}$ for the GLASS-AVHRR product. This variability may
be attributed to the different footprint radii of COSMOS sensors, which result in varying degrees of spatial
representativeness. Besides, the biases of the GLASS-AVHRR SM product across the COSMOS networks were much larger
than those observed across the ISMN network, particularly on the COSMOS-UK network, where the median bias reached -
0.09 $m^3$ $m^{-3}$. This is likely due to the considerably greater sensing depth of the COSMOS sensors (15–83 cm), compared to
the GLASS-AVHRR SM product (up to 5 cm). Meanwhile, although the first-layer (0–7 cm) ERA5-Land SM product was
used here for evaluation, it still exhibited large wet biases across these COSMOS networks, further suggesting its extensive
overestimation issue.




**Figure 6** Boxplots of R, bias, and ubRMSE for the GLASS-AVHRR SM product across different groups of ISMN stations and three field-scale COSMOS networks, in comparison with the GLASS-MODIS and ERA5-Land SM products. The number above each box represents the median value of the metrics across all stations within each network.

Despite the high accuracy achieved when validating the GLASS-AVHRR SM product using both the point-scale ISMN and field-scale COSMOS in situ SM datasets, the validation results were inevitably affected by the scale differences between these datasets. Therefore, the upscaled 9 km SMAP CVS in situ SM dataset from 22 different locations was also utilized to validate the GLASS-AVHRR SM product from 2015 to 2021 as a complement. Specifically, the mean SM values of the 5 km GLASS-AVHRR SM product within a 2 × 2 window corresponding to each 9 km SMAP CVS grid were first calculated, and then the validation metrics for the GLASS-AVHRR SM product were estimated at each CVS, as listed in Table 6. As a

comparison, validation metrics for the ERA5-Land SM product (~ 9 km horizontal resolution) were also calculated at each CVS and presented in the table.

**Table 6** Validation metrics for the GLASS-AVHRR and ERA5-Land SM products at 22 upscaled 9 km SMAP core validation sites.

| Site | GLASS-AVHRR | | | | ERA5-Land | | | | LC | No. |
| | R | bias ($m^3$ $m^{-3}$) | RMSE ($m^3$ $m^{-3}$) | ubRMSE ($m^3$ $m^{-3}$) | R | Bias ($m^3$ $m^{-3}$) | RMSE ($m^3$ $m^{-3}$) | ubRMSE ($m^3$ $m^{-3}$) | | |
|---|---|---|---|---|---|---|---|---|---|---|
| HOBE | 0.61 | -0.07 | 0.100 | 0.069 | 0.63 | -0.02 | 0.069 | 0.066 | Croplands | 252 |
| Kenaston1 | 0.76 | -0.07 | 0.078 | 0.036 | 0.72 | 0.02 | 0.051 | 0.048 | Croplands | 87 |
| Kenaston2 | 0.80 | -0.08 | 0.084 | 0.035 | 0.77 | 0.01 | 0.046 | 0.045 | | 87 |
| Carman | 0.71 | 0.01 | 0.042 | 0.042 | 0.61 | 0.10 | 0.115 | 0.053 | Croplands | 145 |
| South Fork | 0.61 | 0.00 | 0.062 | 0.062 | 0.67 | 0.07 | 0.096 | 0.060 | Croplands | 179 |
| St. Josephs | 0.71 | -0.07 | 0.077 | 0.037 | 0.75 | 0.05 | 0.063 | 0.035 | Croplands | 115 |
| REMEDHUS1 | 0.87 | 0.05 | 0.051 | 0.022 | 0.86 | 0.16 | 0.172 | 0.071 | Croplands | 557 |
| REMEDHUS2 | 0.86 | -0.04 | 0.050 | 0.034 | 0.84 | 0.09 | 0.101 | 0.046 | | 540 |
| Valencia | 0.54 | -0.01 | 0.047 | 0.045 | 0.59 | 0.08 | 0.111 | 0.078 | Savannas | 107 |
| Tonzi Ranch | 0.95 | 0.00 | 0.030 | 0.030 | 0.94 | 0.09 | 0.097 | 0.045 | Savannas | 79 |
| Fort Cobb | 0.81 | 0.01 | 0.034 | 0.034 | 0.83 | 0.08 | 0.085 | 0.040 | Grasslands | 248 |
| Little Washita | 0.78 | 0.01 | 0.039 | 0.038 | 0.77 | 0.05 | 0.071 | 0.049 | Grasslands | 225 |
| Walnut Gulch1 | 0.71 | 0.01 | 0.030 | 0.027 | 0.69 | 0.01 | 0.062 | 0.061 | Shrublands | 159 |
| Walnut Gulch2 | 0.74 | 0.04 | 0.042 | 0.021 | 0.71 | 0.11 | 0.126 | 0.062 | | 189 |
| Little River | 0.36 | 0.00 | 0.043 | 0.043 | 0.76 | 0.22 | 0.225 | 0.040 | Cropland/ Natural mosaic | 84 |
| TxSON1 | 0.87 | 0.00 | 0.024 | 0.024 | 0.88 | 0.09 | 0.100 | 0.040 | Grasslands | 55 |

Earth System
Science
Data

| | | | | | | | | | | |
|---|---|---|---|---|---|---|---|---|---|---|
| TxSON2 | 0.90 | 0.02 | 0.028 | 0.023 | 0.91 | 0.07 | 0.076 | 0.038 | | 103 |
| Niger | 0.73 | 0.00 | 0.018 | 0.017 | 0.69 | 0.04 | 0.061 | 0.046 | Grasslands | 138 |
| Benin | 0.91 | 0.04 | 0.052 | 0.037 | 0.88 | 0.22 | 0.228 | 0.062 | Savannas | 217 |
| Monte Buey | 0.78 | -0.07 | 0.081 | 0.035 | 0.74 | 0.01 | 0.053 | 0.052 | Croplands | 120 |
| Yanco1 | 0.92 | -0.02 | 0.049 | 0.043 | 0.87 | 0.04 | 0.064 | 0.050 | Croplands | 121 |
| Yanco2 | 0.90 | 0.00 | 0.035 | 0.035 | 0.86 | 0.09 | 0.095 | 0.041 | Grasslands | 117 |
| Average | 0.77 | -0.01 | 0.050 | 0.037 | 0.77 | 0.08 | 0.099 | 0.053 | / | / |
| All | 0.82 | -0.01 | 0.054 | 0.054 | 0.65 | 0.09 | 0.119 | 0.083 | / | 3924 |



**Figure 7** Scatter plots between the upscaled in situ SM and the corresponding estimated SM from the GLASS-AVHRR or ERA5-Land product at each SMAP core validation site.

At most of the CVSs, the GLASS-AVHRR SM product achieved similar R values to the ERA5-Land SM product, except at the Little River site, where the R value for the GLASS-AVHRR product was significantly lower. This is probably because the land cover type at this site is "Cropland or Natural mosaic", making the upscaled in situ SM measurements less





representative. Meanwhile, while the GLASS-AVHRR SM product exhibited notable dry biases only at a few CVSs, the ERA5-Land SM product showed large wet biases at most of the CVSs, as also reported in detail by Lal et al. (2022). The varying degrees of bias in these two SM products can be more intuitively observed through their scatter plots against the upscaled in situ SM at each CVS (Fig. 7). As one of the main inputs for generating the GLASS-AVHRR SM product, the ERA5-Land reanalysis SM exhibited notable wet biases at almost all CVSs, especially at REMEDHUS1, Little River, and

Benin, which were largely corrected by the GLASS-AVHRR product, with the data points on the scatter plots being much closer to the 1:1 line. This can be attributed to the use of the GLASS-MODIS SM product as training target, although it may have also contributed to the slight dry bias in the GLASS-AVHRR SM product, given that optical and thermal satellite SM estimates typically represent a shallower depth than in situ SM datasets. In addition, at the CVS where the ERA5-Land product exhibited a large wet bias, the RMSE and ubRMSE values of the GLASS-AVHRR product were often much lower

than those of the ERA5-Land product. The average R and ubRMSE values achieved by the GLASS-AVHRR SM product at 22 CVSs were 0.77 and 0.037 $m^3$ $m^{-3}$, respectively, similar to those reported for the 9 km SMAP-Sentinel L2 SM product, which were 0.79 and 0.035 $m^3$ $m^{-3}$, respectively (Das et al., 2020). When combining all the CVS in situ SM measurements, an overall R of 0.82 and ubRMSE of 0.054 $m^3$ $m^{-3}$ were obtained by the GLASS-AVHRR SM product, showing significant improvement over the ERA5-Land SM product, which had values of 0.65 and 0.083 $m^3$ $m^{-3}$, respectively. This is also

evident from the more concentrated scatter points of the GLASS-AVHRR SM product displayed in Fig. 7.

To intuitively examine the ability of the GLASS-AVHRR SM product to capture temporal variations in measured SM and its temporal consistency with the GLASS-MODIS product, time-series curves for the GLASS-AVHRR (aggregated at 10 km), GLASS-MODIS (aggregated at 9 km), and in situ SM (upscaled at 9 km) at six CVSs with different land cover types were plotted, with the ERA5-Land SM product (~ 9 km horizontal resolution) also included for reference (Fig. 8). Through

extending the GLASS-MODIS SM product from 2000 back to 1982, the GLASS-AVHRR SM product attained complete temporal coverage from 1982 to 2021, and a high degree of temporal consistency between these two products could be observed from the time-series plots. Despite that the ERA5-Land SM product also had long-term temporal coverage, it exhibited large wet biases when compared with the upscaled in situ SM at all six CVSs, whereas both the GLASS-MODIS and GLASS-AVHRR SM products aligned more closely with the dynamic ranges of measured SM. As mentioned above, the

GLASS-AVHRR SM product exhibited notable dry biases at a few CVSs. However, as can be seen from the time-series curves at REMEDHUS2 (Fig. 8a) and Yanco1 (Fig. 8f), suspicious abrupt rises in measured SM, as well as temporary spikes in SM (possibly caused by irrigation), might also have partially contributed to these dry biases. Overall, the GLASS-AVHRR SM product could well capture the temporal variations in measured SM at these CVSs, except for the Little River site (Fig. 8d) where the land cover type is "Cropland or Natural mosaic". Measured SM at this site did not show a clear

seasonal pattern as at the other sites, and there was less consistency between the two GLASS SM products, likely due to the stronger spatial heterogeneity of this site. Besides, at the Walnut Gulch1 site (Fig. 8c) where the dominant land cover type is "shrubland", while the GLASS-AVHRR product captured high SM values well, it slightly overestimated when the measure SM approached zero.









**Figure 8** Time-series plots of the GLASS-AVHRR (aggregated at 10 km), GLASS-MODIS (aggregated at 9 km), ERA5-Land (~ 9 km horizontal resolution), and in situ SM (upscaled at 9 km) at six CVSs with different land cover types for the period 1982–2021.

### 4.3 Spatial consistency with global SM products

To further investigate the spatial consistency between the GLASS-AVHRR and GLASS-MODIS SM products, as well as
with two widely used long-term global SM products, mean SM maps of the GLASS-AVHRR, GLASS-MODIS, ESA CCI, and ERA5-Land products were plotted for January and July of 2016, respectively (Fig. 9). It can be seen that the GLASS-AVHRR SM product had the most complete spatial coverage among these products, after masking out permanent snow and ice and water bodies (Fig. 9g-h). Despite the spatiotemporal continuity of the ERA5-Land reanalysis SM product, it yielded negative SM values close to zero in parts of the northern Africa, especially in July, which were masked out here (Fig. 9c-d).
The ESA CCI combined SM product exhibited substantial spatial gaps above 30° N in January, in addition to the persistent absence of valid estimates in some densely vegetated regions (e.g., the Congo River and Amazon River basins), due to the attenuation of microwave signals in these areas (Fig. 9a-b) (Dorigo et al., 2017). Meanwhile, because of the lack of GLASS-MODIS albedo products at high latitudes during the cold season, GLASS-MODIS SM estimates were unavailable at high latitudes (above 60° N) in January (Fig. 9e). Nevertheless, this does not affect the complete spatial coverage of the GLASS-
AVHRR SM product, although it should still be used with caution when LST is below zero degree Celsius.

In terms of the spatial distribution patterns of SM, the GLASS-AVHRR and GLASS-MODIS SM products showed a high degree of consistency, which further demonstrates the effectiveness of the developed DL model. In general, both GLASS SM products were slightly drier than the ESA CCI combined SM product, probably because optical and thermal satellite SM estimates typically represent a shallower depth compared to microwave SM products. In contrast, the ERA5-Land SM
product was much wetter than the other three SM products, especially in regions with high SM levels. While the three satellite SM products generally ranged between 0 and 0.5 $m^3$ $m^{-3}$, the ERA5-Land reanalysis SM product showed a range of 0–0.7 $m^3$ $m^{-3}$, indicating a clear tendency for overestimation. Although varying degrees of biases existed among the four global SM products, similar spatial patterns could be observed in all of them, characterized by higher SM values in the eastern United States, northern South America, central Africa, and southern Asia, and lower SM values in the western USA,
Middle East, northern and southern Africa, and Australia. Moreover, July was slightly drier than January in all four SM products, particularly in regions such as the western USA, eastern South America, and central Asia.





**Figure 9** Mean global SM maps of the (a–b) 0.25° ESA CCI combined, (c–d) 0.1° ERA5-Land, (e–f) 1 km GLASS-MODIS, and (g–h) 5 km GLASS-AVHRR SM product in January and July of 2016.

Figure 10 presents a zoomed-in comparison between the four SM products across the Tibetan Plateau in July 2016. The Tibetan Plateau, located in Central Asia, is the highest and most extensive plateau in the world, with an average elevation exceeding 4,000 meters. Its climate is extreme and varied, featuring significant seasonal and interannual variations. The unique topographic and climatic characteristics of the Tibetan Plateau make it one of the hotspots for global climate change research. As can be observed from Fig. 10, all of the SM products show similar spatial distribution patterns: lower SM levels

in the western and northern parts of the plateau, where rainfall is scarce and vegetation is sparse, and higher SM levels in the

eastern and southern regions, where rainfall is more abundant and vegetation is denser. The GLASS-AVHRR SM product also exhibited high spatial consistency with the GLASS-MODIS SM product over the Tibetan Plateau, indicating that the adopted DL model effectively learned spatial features from the target SM product without introducing significant biases. Compared to the other three products, the ERA5-land SM product was much wetter in the southern part of the plateau, and the large positive bias in the ERA5-land reanalysis SM over the Tibetan Plateau was also reported in a previous study (Xing et al., 2021). Notably, there were many small patches with abrupt SM changes in the ERA5-land product (Fig. 10c), which were markedly improved in both the GLASS-AVHRR and GLASS-MODIS SM products. Moreover, compared to the ERA5-land and ESA CCI SM products at coarser resolutions, the GLASS-AVHRR SM product contained much richer spatial details and could well capture the distribution patterns of topography and vegetation.

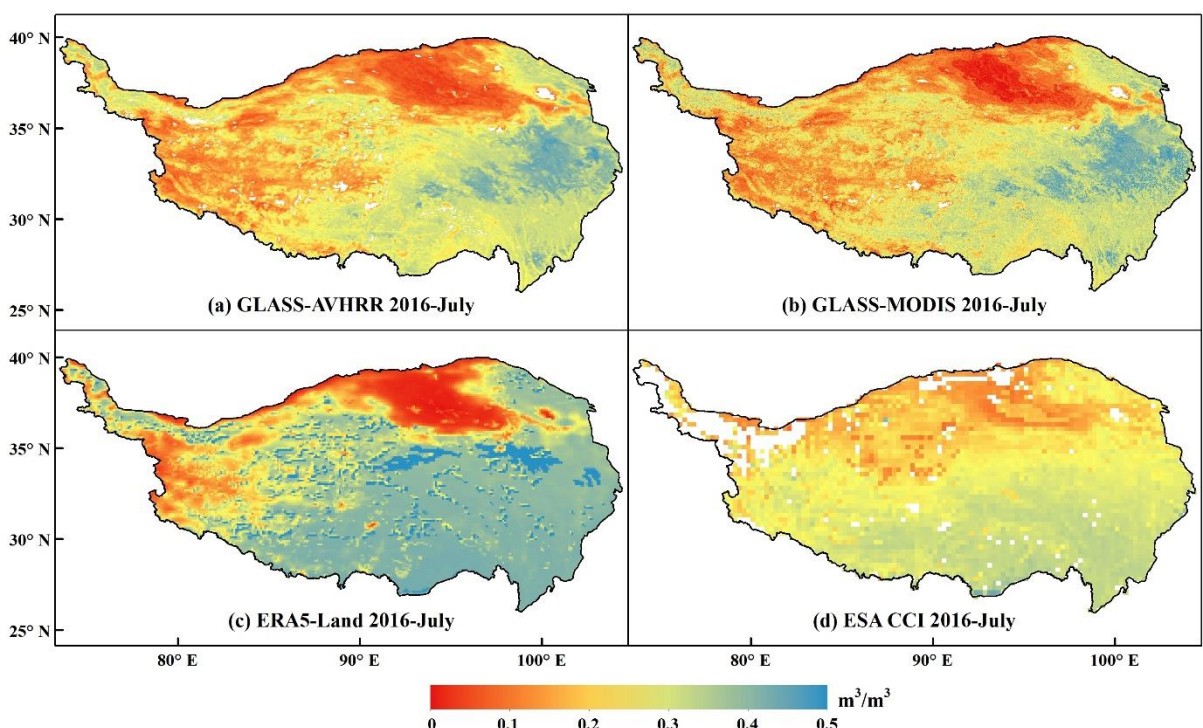

**Figure 10** Zoomed-in comparison of the (a) 5 km GLASS-AVHRR, (b) 1 km GLASS-MODIS, (c) 0.1° ERA5-Land, and (d) 0.25° ESA CCI combined SM products across the Tibetan Plateau in July 2016.

**5 Discussion**

This study aimed to develop a long-term global SM estimation model based on deep learning, so as to derive a temporally consistent SM product with reliable accuracy over the last four decades. Therefore, we mainly explored two types of widely used DL models that are adept at processing sequential data: the LSTM-based models and Transformer. While LSTM has





been utilized to retrieve SM since 2017 (Fang et al., 2017), the state-of-the-art Transformer model is still rarely used for SM estimation. Specifically, the accuracy of these DL models was compared from multiple perspectives, such as comparisons
between the DL models and benchmark tree-based XGBoost model, between models with different attention mechanisms or different application architectures. Results showed that the Attention-based LSTM (AtLSTM) model achieved the best performance on the test set, and the MTM architecture could output a sequence of SM estimates simultaneously while maintaining similar accuracy to that of the MTO architecture. Note that, Transformer was reported to outperform LSTM-based models in several hydrological applications due to its ability to better handle long sequences and relate any two
positions in the sequence (Amanambu et al., 2022; Yin et al., 2022), and according to Xu et al. (2021), it achieved similar accuracy to the AtLSTM model in multi-temporal crop mapping tasks. However, Zeng et al. (2023) found that a simple linear model can outperform Transformer in long-term time-series forecasting tasks, and ascribed this to the temporal information loss associated with the self-attention mechanism. Thus, the suitability of Transformer for time-series forecasting or estimating remains a topic of ongoing debate. In our study, the accuracy of the Transformer model was
slightly lower than that of the AtLSTM model, particularly for samples with high SM levels ($> 0.4$ $m^3$ $m^{-3}$). Given the high temporal variability of SM and the relatively short temporal length of SM memory, which typically ranges from 5 to 40 days and diminishes with increasing time lags (Orth and Seneviratne, 2012), this result may be attributed to the superior ability of the AtLSTM model to capture short-term adjacent dependencies. Yet, additional experiments with diverse training datasets are necessary to confirm the general applicability of this result.

We also investigated the effect of input sequence length on model accuracy, and it was found that the overall accuracy of the AtLSTM model with the MTM architecture leveled off at a sequence length of about 4 days. Subsequent analysis of the distribution of attention weights indicated that the model could automatically learn the necessary temporal information from adjacent positions in the sequence to accurately estimate SM. Despite that the overall accuracy of the LSTM-based models with the MTM architecture would converge as long as the length of the input sequence is sufficiently long, the models'
accuracy is typically lower at the beginning or end of the sequence, and the affected estimates need to be identified and removed. In contrast, most of the current LSTM or Transformer application architecture is MTO and their accuracy remains unaffected at both ends of the sequence. But it is still necessary to identify the optimal sequence length during the training process to improve model efficiency, as the amount of input data would increase substantially with increasing sequence length. Here, we mainly explored the ability of the LSTM-based models and Transformer to capture temporal information
from time-series input datasets for SM estimation. Future research could consider incorporating spatial patterns by combining the AtLSTM or Transformer models with CNNs, or adapting the network of Transformer to improve its applicability for time-series estimating tasks. Moreover, different input features and data sources can also be integrated to investigate whether the estimation accuracy of SM can be further improved.

To examine the accuracy and consistency of the generated four-decade global daily GLASS-AVHRR SM product, different
strategies were combined to fully evaluate it, including the validation against in situ SM datasets from point-scale ISMN stations, field-scale COSMOS networks, and upscaled 9 km SMAP CVSs, respectively, as well as the intercomparison with

two widely used long-term global SM products. However, the evaluation of the GLASS-AVHRR SM product is still subject to certain limitations. The ISMN in situ SM dataset prior to 2000 is relatively scarce, with only 45 independent stations available for evaluation during this period, and large scale difference exists between this point-scale SM dataset and the 5 km GLASS-AVHRR SM product. The COSMOS sensors generally have varying footprint radii and sensing depths, which can lead to the spatial and vertical representativeness issues. Additionally, there is only a limited number of upscaled SMAP CVSs, and the data collected may also contain errors caused by varying degrees of spatial representativeness. Although validation results showed that high accuracy was achieved by the GLASS-AVHRR SM product at different spatial scales, its accuracy was inevitably influenced by the GLASS-MODIS SM product, which was used as the training target for the corresponding SM estimation model. Therefore, more representative long-term in situ SM datasets are needed to better validate and further improve the quality of the long-term global SM product.

Intercomparison with the long-term ERA5-Land and ESA CCI combined SM products showed that the derived GLASS-AVHRR SM product achieved the most complete spatial coverage, contained much richer spatial details, and remained unaffected by the large wet biases present in the input ERA5-Land SM product. Since no single microwave sensor covered the sufficiently long time period (> 30 years) required for a climate data record, SM products retrieved from multiple sensors using different algorithms were synthesized to generate the ESA CCI combined SM product, which also led to variations in its accuracy over time and space (Dorigo et al., 2012). In contrast, the GLASS-AVHRR SM product was estimated using mainly the seamless GLASS-AVHRR albedo and LST products retrieved from the long-archived AVHRR satellite observations spanning four decades, which ensured its spatial and temporal completeness and consistency. Moreover, although microwave sensors are more sensitive to SM, their signal are significantly attenuated in densely vegetated areas, resulting in persistent data gaps in the ESA CCI product. While the GLASS-AVHRR SM product is less accurate in these regions (with a median R of 0.57 at 20 COSMOS forest stations), it can provide a valuable complement to microwave SM products. Nevertheless, greater efforts should be devoted to both the development and validation of long-term SM climate data records, and it is also crucial to assess the long-term trends in these SM datasets.

**6 Data availability**

The seamless global 5 km SM product (GLASS-AVHRR SM) at daily scale from 1982 to 2021 is freely accessible at https://glass.hku.hk/casual/GLASS_AVHRR_SM/. Additionally, the annual average GLASS-AVHRR SM dataset was also generated, which can be downloaded from https://doi.org/10.5281/zenodo.14198201 (Zhang et al., 2024). Note that this product represents the volumetric water content in the uppermost soil layer (0–5 cm), with areas of permanent snow and ice and water bodies masked.



# 7 Conclusions

A four-decade (1982–2021) seamless global SM product at 5 km resolution was derived here, denoted as the GLASS-AVHRR SM product. This product was estimated using mainly the long-archived AVHRR satellite observations and multi-source datasets based on deep learning. Specifically, a large number of evenly distributed training samples extracted from the global 1 km daily GLASS-MODIS SM product were used as target to train three LSTM-based models (LSTM, Bi-LSTM, and AtLSTM) and a Transformer model, with an XGBoost model employed as the benchmark. After identifying the AtLSTM as the best-performing model, it was ultimately adopted to generate the long-term GLASS-AVHRR SM product, which was then fully evaluated for reliability and consistency. The main results are summarized as follows:

(1) Evaluation of the models on the test set showed that all four DL models outperformed the benchmark XGBoost model, particularly at high SM levels ($> 0.4$ m$^3$ m$^{-3}$). Notably, the AtLSTM model achieved the best performance, with an $R^2$ of 0.987 and RMSE of 0.011 m$^3$ m$^{-3}$, and its SM estimates were much closer to 1:1 line than those from other models. These results indicate that utilizing bidirectional temporal information from the input sequence as well as adding an attention module are both effective in improving the estimation accuracy of SM. Meanwhile, The MTM architecture adopted in this study achieved similar accuracy to that of the MTO architecture, while being able to output a sequence of SM estimates simultaneously and reduce considerable production time.

(2) The AtLSTM model with the MTM architecture was then employed to investigate the effect of input sequence length on model accuracy, and it was found that the overall accuracy of the model leveled off at a sequence length of about 4 days. Further analysis of attention weights revealed that the AtLSTM model with the MTM architecture could automatically learn the necessary information from adjacent positions in the sequence to accurately estimate SM at each position. In contrast, the temporal information learned by the self-attention module of the Transformer model was more dispersed distributed, and the slightly lower accuracy of the Transformer model than the AtLSTM model might be attributed to the typically high temporal variability of SM and that short-term adjacent temporal information played a more critical role in the accurate estimation of SM.

(3) The derived GLASS-AVHRR SM product was first evaluated using 45 independent point-scale ISMN stations prior to 2000, resulting in a median R of 0.73 and ubRMSE of 0.041 m$^3$ m$^{-3}$. Then, the product was validated against SM datasets from three post-2000 field-scale COSMOS networks, with median R values ranging from 0.63 to 0.79 and median ubRMSE values between 0.044 and 0.065 m$^3$ m$^{-3}$. Validation of the GLASS-AVHRR SM product at 22 upscaled 9 km SMAP CVSs yielded an overall R of 0.82 and ubRMSE of 0.054 m$^3$ m$^{-3}$. Whereas the ERA5-Land SM product had large wet biases at most of the CVSs, the GLASS-AVHRR SM product basically corrected these biases. Moreover, the time-series plots at six CVSs further demonstrated that the GLASS-AVHRR SM product could well capture the temporal variations in measured SM and showed a high degree of temporal consistency with the GLASS-MODIS SM product.



(4) Finally, the GLASS-AVHRR SM product was intercompared with two widely used long-term global SM products to investigate their spatial consistency. With the most complete spatial coverage, the GLASS-AVHRR SM product was slightly drier than the ESA CCI combined SM product, possibly due to the shallower depth it represents, whereas the ERA5-Land SM product exhibited a clear tendency for overestimation. While similar spatial patterns of SM could be observed in all of these products, the GLASS-AVHRR SM product contained much richer spatial details than the two long-term SM products at coarser resolutions.

Our study demonstrates the feasibility of utilizing the attention-based DL model and AVHRR satellite observations to generate long-term global SM product. The derived GLASS-AVHRR SM product has the advantages of long-term coverage, spatial and temporal integrity, reliable accuracy and consistency. As a reliable extension of the GLASS-MODIS SM product and a valuable complement to microwave SM products, this four-decade global SM product will be beneficial for a range of large-scale climate change-related research. Future studies could combine other DL models or integrate different data sources to further improve the quality of the long-term SM product.

**Appendix A: Supplementary figures**

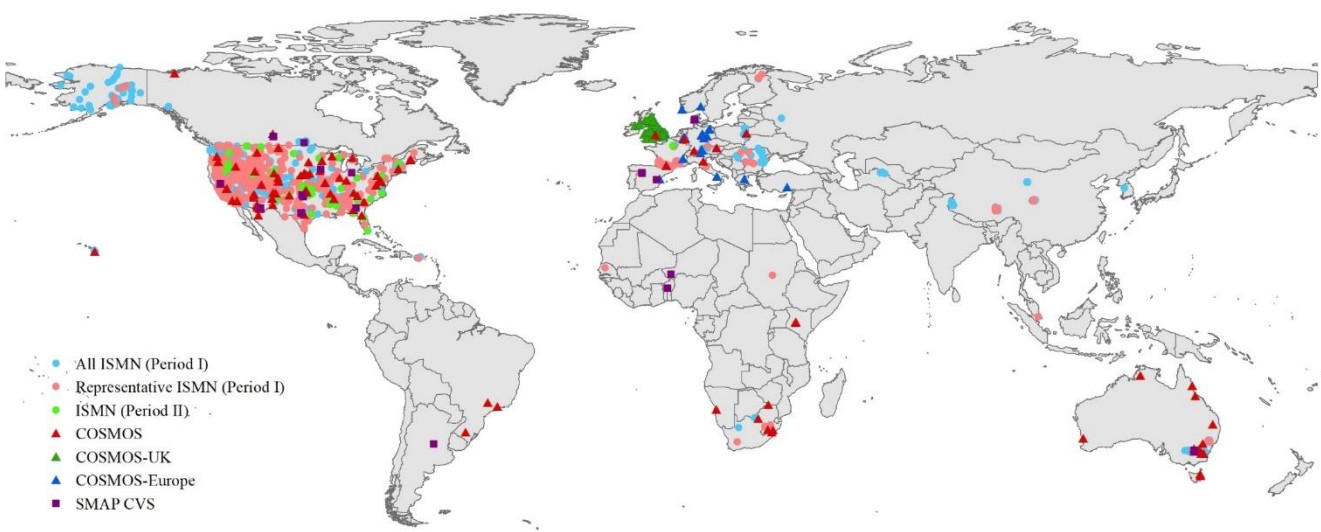

**Figure A1** The spatial distribution of SM stations for each in situ SM dataset used in this study. Period I refers to 2000–2018 and Period II refers to 1982–1999



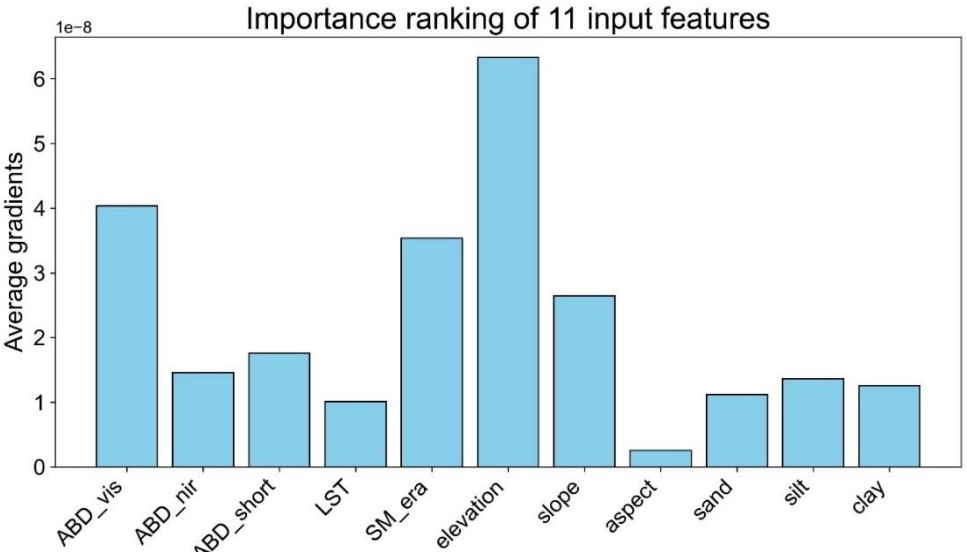

**Figure A2** Importance ranking of 11 input features for the AtLSTM model based on gradients analysis

**Author contributions**

SL and YZ developed the methodology and designed the experiments. YZ, HM, JX, and GZ collected and processed the data. YZ carried out the experiments. TH and FT provided guidance on data analysis and experimental design refinements. YZ prepared the manuscript with contributions from all co-authors.

**Competing interests**

Some authors are members of the editorial board of Earth System Science Data.

**Acknowledgements**

This study was supported by the Open Research Program of the International Research Center of Big Data for Sustainable Development Goals (NO. CBAS2022ORP01), the National Key Research and Development Program of China (2023YFF1303702), the Fundamental Research Funds for the Central Universities, the National Key Research and Development Program of China (NO. 2016YFA0600103), and the National Natural Science Foundation of China (NO. 42090011). We also acknowledge the data support from "National Earth System Science Data Center, National Science & Technology Infrastructure of China (http://www.geodata.cn)".



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
