# Peer review of "A seamless global daily 5 km soil moisture product from 1982 to 2021 using AVHRR satellite data and an attention-based deep learning model"

_Earth System Science Data, 2024_

## Author Comment (AC1)

**CC2: 'Comment on essd-2024-553', SHAOBO SUN**

**General comments**

The authors developed a consistent and seamless global soil moisture product over 1982-2021 using deep learning models and existing SM and auxiliary data. Validation showed that the SM data well capture temporal variability of global SM. Thus, the newly developed, long-term data show potential in wide applications. While, the manuscript needs to be improved.

We sincerely thank you for the positive and encouraging comments on our dataset and its potential applications. We also appreciate the suggestion to improve the manuscript. In response, we have revised the text to enhance its clarity and overall quality. Detailed responses and corresponding changes are provided below.

**Specific comments**

1. The authors did not use SM measurements to establish a DL model and develop global SM product. I think the developed GLASS-AVHRR SM product is a reprocessed SM product from ERA-Land SM and previous GLASS MODIS SM products. In the abstract, the authors said "we generated a consistent and seamless global SM product from 1982 to 2021 using deep learning (DL) by integrating four decades of Advanced Very High Resolution Radiometer (AVHRR) albedo and land surface temperature products with multi-source datasets."; did not mention use of ERA-Land SM which is much important for generating the SM data (Fig. A2). I think the authors should clearly stated how they generated the GLASS-AVHRR SM product with existing datasets in Abstract.

Thank you for pointing out the lack of clarity regarding the data sources used in generating the GLASS-AVHRR SM product. We have revised the abstract to more explicitly state the role of ERA5-Land SM and clarify the datasets used. The updated sentence in the abstract now reads:

"Here, we generated a consistent and seamless global surface SM product (0–5 cm) spanning 1982 to 2021 using a deep learning (DL) model. The model was trained with the GLASS-MODIS SM product and was designed to integrate four decades of Advanced Very High Resolution Radiometer (AVHRR)-derived albedo and land surface temperature, ERA5-Land SM, as well as terrain and soil texture datasets as input features."

2. Page 3: "including land cover mapping, data fusion and downscaling, and environmental parameter retrieval (Yuan et al., 2020). " Please add citations for each

research area.

We have added citations as suggested. In particular, we now cite Huang et al. (2018) for land cover mapping, Wang et al. (2021) for data fusion and downscaling, and Ma and Liang (2022) for environmental parameter retrieval.

Huang, B., Zhao, B., and Song, Y.: Urban land-use mapping using a deep convolutional neural network with high spatial resolution multispectral remote sensing imagery, Remote Sens. Environ., 214, 73–86, https://doi.org/https://doi.org/10.1016/j.rse.2018.04.050, 2018.

Wang, F., Tian, D., Lowe, L., Kalin, L., and Lehrter, J.: Deep Learning for Daily Precipitation and Temperature Downscaling, Water Resour. Res., 57, e2020WR029308, https://doi.org/https://doi.org/10.1029/2020WR029308, 2021.

Ma, H. and Liang, S.: Development of the GLASS 250-m leaf area index product (version 6) from MODIS data using the bidirectional LSTM deep learning model, Remote Sens. Environ., 273, 112985, https://doi.org/10.1016/J.RSE.2022.112985, 2022.

3. Table 1: Many other available long-term SM products were not included, such as the GLEAM SM.

Thank you for the suggestion. We have added two model-based long-term global SM products—GLEAM4 and SiTHv2—in Table 1. In addition, a brief description of these products has been added to the manuscript as follows:

"Recently, models that focus on the dynamic simulation of evapotranspiration and SM, such as the fourth generation of the Global Land Evaporation Amsterdam Model (GLEAM4) (0.1°, 1980–2023) and the Simple Terrestrial Hydrosphere model, version 2 (SiTHv2) (0.1°, 1982–2020), have also provided long-term global SM products by integrating multi-source satellite data and hydrometeorological variables (Miralles et al., 2025; Zhang et al., 2024a)"

Miralles, D. G., Bonte, O., Koppa, A., Baez-Villanueva, O. M., Tronquo, E., Zhong, F., Beck, H. E., Hulsman, P., Dorigo, W., Verhoest, N. E. C., and Haghdoost, S.: GLEAM4: global land evaporation and soil moisture dataset at 0.1° resolution from 1980 to near present, Sci. Data, 12, 416, https://doi.org/10.1038/s41597-025-04610-y, 2025.

Zhang, K., Chen, H., Ma, N., Shang, S., Wang, Y., Xu, Q., and Zhu, G.: A global dataset of terrestrial evapotranspiration and soil moisture dynamics from 1982 to 2020, Sci. Data, 11, 445, https://doi.org/10.1038/s41597-024-03271-7, 2024a.

4. Section 2 datasets: introductions on the datasets used in this section were verbose. I suggest the author simply simplify this section.

Thank you for the suggestion. We have simplified Section 2 by removing some less important sentences to improve clarity and conciseness.

5. Section 3 methods: please also simplify introductions on the DL/machine learning models, and pay much attentions on discussing results.

We have also simplified Section 3 by removing some less important sentences to improve clarity and conciseness.

6. Figure 9: In northern high latitudes, permafrost, snow and ice distribute widely, SM values in these regions are invalid in non-growing seasons. Thus, the SM maps in Jan. should masked these regions.

Thank you for the comment. Figure 9 is intended to present the spatial distribution of various SM products in their original forms for a direct visual comparison. Regions with permanent snow and ice as well as water bodies, where SM values are invalid, have now been masked out in all SM products using the MODIS land cover type product (MCD12C1).

However, masking regions covered by permafrost as well as seasonal snow and ice in the January SM maps would require reliable monthly land cover datasets, which are currently difficult to obtain. Therefore, we have clarified in the figure caption that "SM values in northern high latitudes in January should be interpreted with caution due to the widespread presence of permafrost, snow, and ice."

7. Line 615: "model based on deep learning" - using DL models

Revised as suggested.

8. Page 28: "a topic of ongoing debate" - citations are needed.

**Citations are added as suggested.**

9. Discussion: I suggest the authors pay more attention on discussing their results, including uncertainties in the their developed SM data, and importance of the input variables (Fig. 2A). Specially, Fig. 2A shows that elevation exhibits largest influence on predicting SM. While, the ERA-Land SM had small important value. Why?

Thank you for raising this important point. In response, we have added a discussion on the uncertainties associated with our developed SM product in the Discussion section (Section 5). Specifically, we included the following statement:

"Meanwhile, as a data-driven product, the quality of the GLASS-AVHRR SM product largely depends on the selected input features, their accuracy and consistency, and the representativeness of the training data. Potential uncertainties may arise from biases or errors in the satellite and reanalysis inputs. In particular, the reduced model accuracy observed in the high SM range is likely due to the inherent imbalance in the

numerical distribution of SM samples and increased uncertainty in the accuracy of input features under wet surface conditions. In terms of feature selection, due to constraints such as record length, spatiotemporal completeness, and accuracy requirements, some informative but less consistently available variables may have been excluded, further contributing to the uncertainties in the final SM product. Moreover, as the ERA5-Land reanalysis SM was used as one of the input features, the generated product cannot be considered entirely independent. Future research could explore developing a fully independent, long-term, and seamless global SM product with sufficiently reliable accuracy."

Regarding the importance of input features, we have added the following explanation to the manuscript to clarify the observed differences:

"In particular, although elevation is a static variable, it plays a critical role in shaping the spatial distribution of SM by influencing precipitation, temperature, vegetation type, and evaporation processes. Its impact on the spatial variability of SM tends to be more stable and consistent over time. In contrast, the contributions of dynamic input features such as ERA5-Land SM may fluctuate across time and space and can be diminished by inherent uncertainties and biases in the input data. Moreover, their importance may also be influenced by correlations with other input features."

**RC1: 'Comment on essd-2024-553', Anonymous Referee #1**

This study creates a consistent long-term and high-resolution global soil moisture product with comparable accuracy to previous deep learning approaches and demonstrates a clear superiority of the AtLSTM approach over other compared approaches. The findings and datasets are valuable to the community. The reporting is clear. Overall I recommend it for publication after minor revisions. Please see below for comments:

Thank you very much for your positive feedback and valuable suggestions. Your insightful comments have helped us address several aspects of the manuscript that were previously unclear, greatly enhancing its clarity and rigor. We truly appreciate your time and effort in reviewing our work and improving its quality.

1. The depth of the developed dataset should be noted in the abstract.

Thank you for your suggestion. We have now included the depth of the developed dataset in the abstract to provide clearer information.

Specifically, we have revised the abstract to mention that: "Here, we generated a consistent and seamless global surface SM product (0-5 cm) from 1982 to 2021..."

Additionally, the depth of the dataset is also emphasized in other parts of the manuscript to ensure consistency.

2. The training target is a previously developed 1km dataset, and the newly developed dataset mainly provides the advantage of being longer-term and better winter coverage. This would be clearer if the temporal coverage of the Zhang et al. (2023) dataset is stated in p4 lines 117-118.

Thanks for the reminder. We have now added the temporal coverage of the GLASS-MODIS SM product (2000–2020) to distinguish it from the newly developed dataset.

3. p5 line 146: "8-day temporal resolution interpolated to daily" Please specify which interpolation method was applied.

We applied linear interpolation and have emphasized this in the manuscript.

4. It would be nice to add 1-2 sentences about uncertainty in the sampling strategy of the training target (p9 lines 242-p10 ln 244).

We appreciate your advice and have included an analysis of the uncertainty of the sampling strategy, as follows: "While the three years were selected to represent different periods within the available time span (2000–2020), this selection may

introduce some uncertainty, as climate and environmental conditions can vary annually, and extreme weather or climate events in certain years may affect the representativeness of variables such as LST and SM. Nevertheless, this approach was adopted to control the sample size while ensuring the representativeness of samples across different years."

5. p13 Table 4: Should the number of layers be "2" for Bi-LSTM and 1 elsewhere?

Thanks for pointing out this mistake. Upon rechecking the code and experimental notes, we found that the "number of layers" for all DL models is 1. This has been corrected in Table 4.

Following the standard convention in deep learning frameworks (e.g., PyTorch), in our Bi-LSTM model, "Number of layers = 1" refers to a single Bi-LSTM layer, which internally includes both forward and backward LSTM units.

We realized that the relevant sentence in the manuscript caused some confusion and have revised it from "The Bidirectional LSTM (Bi-LSTM) extends the LSTM network by using two separate LSTM layers to process the input sequence from both forward and backward directions, and then concatenating the outputs of both layers" to "The Bidirectional LSTM (Bi-LSTM) extends the LSTM network by incorporating both forward and backward LSTM units within a single layer, allowing the model to capture contextual information from both directions before concatenating their outputs."

6. The sequence lengths in Table 4 span whole year. The testing on p17 only spans 0-29 days and demonstrates stabilized performances at much shorter lengths than 29 days. Why the drastic increase in sequence lengths in production runs? Also, since the input data are 3 discrete years, please specify what values are used to pad the 60 days before the start and after the end of the whole year.

Given the need to generate daily SM products over 40 years, using an input sequence length of 10–30 days per run and generating SM estimates for such short periods would significantly increase the time required for data preprocessing. To balance accuracy and efficiency, we extended the sequence length to 425 days, enabling the generation of a full year's product in a single run.

During both the training and production phases, the first and last 30 days of each 425-day sequence were padded with actual data from adjacent years to ensure consistency. We have clarified this point in the manuscript.

7. p14 lines 366-367: A bit confusing because this reduced-sample testing was not described in the Methods. Could you add a description?

We have added a description of the purpose of this reduced-sample experiment in the

manuscript and revised the relevant sentence as follows: "The fairly high overall accuracy of the benchmark XGBoost model may be attributed to the large number of training samples, specifically 135,360 pixels per day over 3 years, evenly distributed across the globe on a daily basis. To evaluate the impact of sample size on model performance, we conducted an experiment by reducing the number of training samples. When the sample size was reduced by a factor of 100, the accuracy of the XGBoost model dropped considerably, with an R2 of 0.96 and RMSE of 0.017 m3 m-3 on the test set. This highlights the importance of having sufficient samples to achieve high accuracy with XGBoost and indicates the advantage of using the GLASS-MODIS SM product as training target, which can provide much richer samples than the sparse in situ ISMN SM dataset."

8. p15 Table 5: Is there a specific reason for choosing 0.4 m3/m3 as a threshold for large SM values?

The threshold of 0.4 m3 m-3 was chosen because soil moisture values exceeding this level are generally considered high in most global land areas, except for wetland and irrigated regions. This threshold enables a more detailed assessment of model performance under high soil moisture conditions, where sample availability is relatively limited. This pattern can also be observed in the density scatter plots shown in Fig. 3, where the red dots, indicating the highest density, are primarily concentrated within the  $0-0.4 \text{ m}^3 \text{ m}^{-3}$  range.

9. Fig. 6: COSMOS dataset is not directly comparable to the developed dataset due to significant depth differences. It is okay as an auxiliary comparison, given that other comparable independent validation datasets are used (1982-1999 ISMN and SMAP validation sites). However, for informative purpose, it would be good to provide boxplots of the depths of the COSMO observations used in the last three columns of Fig. 6 in supplementary info.

Following your suggestion, we have summarized the sensing depths of all sensors across the three COSMOS networks and added a boxplot (Fig. A2) in the Appendix. We have also included relevant descriptions in the main text.

"The distribution of sensing depths for each station across the three COSMOS networks is presented in Fig. A2. While COSMOS sensors measure SM at relatively deeper layers, they have been used to validate microwave and modelled surface SM products and show good correlations with them (Montzka et al., 2017; Peng et al., 2021b)."

Montzka, C., Bogena, H. R., Zreda, M., Monerris, A., Morrison, R., Muddu, S., and Vereecken, H.: Validation of Spaceborne and Modelled Surface Soil Moisture Products with Cosmic-Ray Neutron

Probes, Remote Sens., 9, https://doi.org/10.3390/rs9020103, 2017.

Peng, J., Tanguy, M., Robinson, E. L., Pinnington, E., Evans, J., Ellis, R., Cooper, E., Hannaford, J., Blyth, E., and Dadson, S.: Estimation and evaluation of high-resolution soil moisture from merged model and Earth observation data in the Great Britain, Remote Sens. Environ., 264, 112610, https://doi.org/https://doi.org/10.1016/j.rse.2021.112610, 2021.

Figure A2 Boxplots of sensing depths across the three COSMOS networks used for validation

10. More on Fig. 6. The currently used 1982-1999 ISMN set provides spatiotemporally independent comparison. The 1982-1999 data at the 715 representative stations would still be temporally independent comparison. Could you include the performance metrics on this subset of data?

Thank you for your suggestion. However, the vast majority of ISMN SM monitoring stations were established after 2000. Among the 715 representative stations, only 23 had observations before 2000. Comparing the metrics obtained from these two groups of datasets may not be sufficient to draw robust conclusions. Therefore, we opted not to include this comparison in Fig. 6. Nevertheless, we assessed the validation metrics of the GLASS-AVHRR SM product across the 23 representative stations from 1982 to 1999, yielding a median R of 0.76, a median bias of -0.03 m3 m-3, and a median ubRMSE of 0.029 m3 m-3.

**RC2: 'Comment on essd-2024-553', Anonymous Referee #2**

This study used several deep learning models to estimate SM with some optical remote sensing indices, soil properties and DEM. Even though it's a big work, it lacks sufficient innovation and employs inappropriate evaluation method. Despite the overall scale of the work, the methodological design raises serious concerns.

We sincerely thank you for your time and effort in reviewing our manuscript and for recognizing the scale of our work. With due respect, we would like to express our disagreement with the comment that the study lacks sufficient innovation and adopts inappropriate evaluation method.

In terms of methodological innovation, our study explored state-of-the-art deep learning models to generate a seamless, four-decade global SM product at 5 km resolution. By integrating multi-source datasets, this product achieves complete spatiotemporal coverage, reliable accuracy, rich spatial details, and low biases. Given the limited research on developing long-term global SM products based on deep learning, along with the notable strengths of our product, we believe this work meets the journal's requirements for innovation and high-quality datasets.

Regarding the evaluation strategy, we carefully designed the validation process by incorporating multiple methods to assess the performance of our model and product. During the model development phase, we used the GLASS-MODIS SM product—also employed as the training target—as a reference to evaluate the model's estimated SM on the test set. The high accuracy achieved indicates the strong agreement between the estimated and target SM values, and demonstrates the reliability of the GLASS-AVHRR product as a temporal extension of the GLASS-MODIS product. During the product accuracy evaluation phase, we fully evaluated the product's accuracy using independent in situ SM datasets across different spatial scales, including point-scale ISMN stations, field-scale COSMOS networks, and 9-km upscaled SMAP core validation site, as detailed in Section 4.2. In addition, time series comparisons and spatial consistency analyses with other long-term global SM products provided further evidence of the robustness and reliability of our product.

We hope these clarifications help to alleviate your concerns and highlight the novel aspects and methodological soundness of our work.

**Major Comments:**

Line 135, Table 2: I question the innovation and scientific contribution of using ERA5-Land SM as an input variable in a model that aims to estimate soil moisture. This approach may introduce circular reasoning and undermines the novelty of the proposed method.

We sincerely appreciate your insightful comment and understand your concern regarding potential circular reasoning—specifically, the risk that using ERA5-Land SM as an input while also estimating SM might lead the model to simply reproduce the ERA5-Land SM, rather than genuinely learning from multi-source datasets.

We would like to clarify that ERA5-Land SM was not used as a target variable, nor was the intention to replicate its pattern. Instead, it served as one of several input features—alongside satellite observations, terrain, and soil properties—aimed at improving the model's ability to capture the complex spatiotemporal variability of SM. Furthermore, our final product was fully validated using independent in situ SM datasets (ISMN, COSMOS, and SMAP CVSs), and the results showed that the biases in our product were significantly lower than those in the ERA5-Land SM product. As such, our approach focuses on multi-source data fusion, leveraging the strengths of various datasets, rather than merely "using SM to predict SM." Therefore, we believe that the use of ERA5-Land SM as an input does not undermine the novelty of our approach or lead to circular reasoning.

Accordingly, we explicitly clarified the objective of this study in the manuscript:

"(1) To develop a DL-based global SM estimation model by integrating multisource datasets and leveraging their complementary strengths, so as to derive a seamless and reliable long-term global SM product."

Line 380, Table 5: It is inappropriate to use GLASS-MODIS SM as a reference to evaluate the model's estimated SM, as they are not independent. Since variables such as LST and albedo were also used in the model estimation, the evaluation becomes biased, leading to inflated performance metrics (e.g.,  $R^2 > 0.98$ ). This dependence compromises the reliability of the validation.

Thank you for pointing out this important concern. We would like to clarify that the use of GLASS-MODIS SM as a reference in Table 5 was solely part of the model development process—specifically, to assess how well the models could reproduce the training target on a held-out test set. This evaluation is a standard practice in deep learning studies to verify model learning behavior before proceeding to independent validation.

Table 5 aims to compare the performance of different DL models on the test set in order to select the model with the highest accuracy. The high R2 values reported here reflect the models' fit to the training target under consistent input settings, but they do

not represent the final product's accuracy. In contrast, the accuracy of our final product was thoroughly evaluated in Section 4.2 using independent in situ SM datasets across different spatial scales.

To avoid any potential misunderstanding that the model's accuracy on the test set represents the actual accuracy of the GLASS-AVHRR SM product, we have removed the reference to model accuracy in the abstract and revised the relevant text as follows: "Our results show that all four DL models outperformed the benchmark XGBoost model, with the AtLSTM model achieving the highest accuracy on the test set, particularly at high SM levels (>  $0.4 \text{ m}^3 \text{ m}^{-3}$ )."

Line 20: The five deep learning models yielded very similar results, with R2 values ranging from 0.982 to 0.987. Therefore, the claim that these models "...effectively enhance..." SM estimation is not well supported.

Thank you for your valuable comments. We fully understand your concern regarding the similarity in overall accuracy among the five models. However, it is important to point out that the AtLSTM model significantly improved SM estimation accuracy under high SM conditions. Specifically, on the test set, the XGBoost model exhibited notable underestimation at high SM levels (> 0.4 m3 m-3), yielding an R2 of 0.413 and RMSE of 0.022 m3 m-3. In contrast, by incorporating both temporal information and attention mechanisms, the AtLSTM model effectively reduced this underestimation and achieved an R2 of 0.621 and RMSE of 0.016 m3 m-3 within the same range, demonstrating superior modeling capability under these challenging conditions.

Therefore, although the overall R2 values across these models appear similar, we believe that this significant improvement at high SM levels by the AtLSTM model supports the claim that temporal and attention-based modeling can effectively enhance the estimation accuracy of SM. To improve clarity, we have revised the relevant statement in the abstract to: "These results suggest that under some challenging conditions, utilizing temporal information as well as adding an attention module can effectively enhance the estimation accuracy of SM."

**Minor Comments:**

Line 60: There are additional remote sensing-based SM products that should be referenced, including but not limited to: Cheng et al. (2023); Guevara, Taufer, & Vargas (2021); and Zheng, Jia, & Zhao (2023).

Thank you for the valuable information. We have added citations to Cheng et al. (2023) and Guevara et al. (2021) in Page 3 of the revised manuscript, respectively. Zheng et al. (2023) had already been cited in the same page.

Line 190: The time period referred to as "period1" is 2000–2018. Why were the years 2019–2021 excluded from the analysis?

The period 2000–2018 was intentionally selected to ensure consistency with the validation period used in our previous paper on the GLASS-MODIS SM product, as illustrated in Fig. 5 of Zhang et al. (2023). In this work, we also used the GLASS-MODIS SM validation results at ISMN stations as a benchmark for comparison. Aligning the time period helps avoid potential confusion for readers regarding any differences in the reported performance of the same product between the two studies.

Zhang, Y., Liang, S., Ma, H., He, T., Wang, Q., Li, B., Xu, J., Zhang, G., Liu, X., and Xiong, C.: Generation of global 1 km daily soil moisture product from 2000 to 2020 using ensemble learning, Earth Syst. Sci. Data, 15, 2055–2079, https://doi.org/10.5194/essd-15-2055-2023, 2023.

Line 355: The statement "two widely used long-term global SM products" should specify which products are being referred to for clarity.

Thank you for the comment. We have revised the sentence to specify the two SM products, namely the ERA5-Land and ESA CCI.

Cheng, F., Zhang, Z., Zhuang, H., Han, J., Luo, Y., Cao, J., . . . Tao, F. (2023). ChinaCropSM1 km: a fine 1 km daily soil moisture dataset for dryland wheat and maize across China during 1993–2018. Earth System Science Data, 15(1), 395-409. doi:10.5194/essd-15-395-2023

Guevara, M., Taufer, M., & Vargas, R. (2021). Gap-free global annual soil moisture: 15 km grids for 1991–2018. Earth System Science Data, 13(4), 1711-1735. doi:10.5194/essd-13-1711-2021

Zheng, C., Jia, L., & Zhao, T. (2023). A 21-year dataset (2000-2020) of gap-free global daily surface soil moisture at 1-km grid resolution. Sci Data, 10(1), 139. doi:10.1038/s41597-023-01991-w

**RC3: 'Comment on essd-2024-553', Anonymous Referee #3**

The authors compared multiple deep learning methods and selected the most suitable approach to generate a 5 km-resolution soil moisture product based on AVHRR data, spanning four decades. Overall, this is a well-written manuscript, and the scope of the study aligns well with the aims and scope of Earth System Science Data. However, the current version of the manuscript appears overly technical in its presentation. The authors should address the following major and specific comments before the manuscript can be considered for publication.

We sincerely thank you for the positive feedback and constructive suggestions. We have revised the manuscript to reduce technical complexity and improve readability. Below are our point-by-point responses to all major and specific comments.

**General comments**

1. While the description of the deep learning methods is sufficiently detailed, the treatment of the predictor variables requires substantial clarification and improvement. It remains unclear how these variables were selected, particularly the rationale for using ERA5-Land data as model inputs. In my view, the relatively strong performance of GLASS-AVHRR in capturing seasonal variations, especially for the R metric, may be partially attributed to this. The authors should consider producing an independently driven soil moisture product, rather than relying on predictors that may introduce redundancy or circular reasoning.

These input variables are widely used in ML and DL-based SM estimation studies. Their selection was guided by prior literature as well as our experience from previous SM product development. Due to practical constraints such as the need for long temporal sequences (40 years), consistent spatiotemporal coverage, and reliable accuracy, some potentially informative variables were excluded.

The ERA5-Land SM product was selected because of its long-term temporal coverage, physical consistency, and relatively high validation accuracy (e.g., in terms of R). Moreover, The ERA5-Land SM product has been widely adopted in previous SM downscaling studies, which supports its credibility and applicability within our modeling framework.

We acknowledge that the relatively strong performance of the GLASS-AVHRR SM product may be partially attributed to the inclusion of ERA5-Land SM as an input. As shown in the feature importance ranking (Fig. A3), ERA5-Land SM contributes significantly to the model's predictions. However, other variables—such as elevation,

black-sky visible albedo, and slope—also play important roles, indicating that the model integrates diverse sources of information beyond any single input.

Our goal was to generate a seamless and reliable long-term global SM product by integrating multi-source datasets and leveraging their complementary strengths. While we acknowledge that some degree of redundancy may exist, the AtLSTM model is capable of effectively handling this through its attention mechanism, which allows it to dynamically focus on the most relevant features over time.

We understand the concern regarding potential circular reasoning—specifically, the risk that using ERA5-Land SM as an input while also estimating SM might lead the model to simply reproduce the ERA5-Land SM, rather than genuinely learning from multi-source datasets. We would like to clarify that ERA5-Land SM was not used as a target variable, nor was the intention to replicate its pattern. Instead, it served as one of several input features—alongside satellite observations, terrain, and soil properties—aimed at improving the model's ability to capture the complex spatiotemporal variability of SM.

We appreciate the suggestion to explore the development of an independently driven SM product and will consider this direction in future work. However, we believe the current design is appropriate for our objective of generating a long-term global SM product with complete spatiotemporal coverage, reliable accuracy, rich spatial details, and low biases.

In response, we have revised the manuscript to provide a clearer explanation of our variable selection rationale and the role of ERA5-Land SM. Specifically, we added the following sentences:

"These input features are widely used in ML and DL-based SM estimation studies."

"In terms of feature selection, due to constraints such as record length, spatiotemporal completeness, and accuracy requirements, some informative but less consistently available variables may have been excluded, further contributing to the uncertainties in the final SM product."

"(1) To develop a DL-based global SM estimation model by integrating multisource datasets and leveraging their complementary strengths, so as to derive a seamless and reliable long-term global SM product;"

"Moreover, as the ERA5-Land reanalysis SM was used as one of the input features, the generated product cannot be considered entirely independent. Future research could explore developing a fully independent, long-term, and seamless global SM product with sufficiently reliable accuracy." 2. I remain concerned about the generalization capability of the developed deep learning model, particularly considering the strong spatial autocorrelation inherent in the current training, validation, and test data split. Furthermore, the results presented in Table 5 indicate that model performance deteriorates when soil moisture exceeds 0.4 m3/m3, which the authors attribute to a lack of training samples. This observation indirectly suggests that the model performs poorly in unseen regions or conditions, highlighting its limited generalization ability. The authors should explore alternative data splitting strategies to rigorously assess the robustness and generalizability of their approach.

We acknowledge your concern regarding the generalization capability of our DL model. In this study, the training, validation, and test datasets were constructed by randomly sampling from 135,360 pixels distributed across the globe, each representing a time-series sample with spatial independent locations. Specifically, these pixels were selected at an interval of five grid cells (i.e., 25 km), ensuring that the spatial distance between any two samples exceeds 25 km. Given that the spatial autocorrelation of soil moisture typically diminishes beyond this scale, the spatial dependence among samples is sufficiently minimized. Therefore, we believe that the current data splitting strategy can ensure a reliable evaluation of the model's generalization performance.

We agree that the model's performance declines when SM exceeds 0.4 m3 m-3. This is mainly due to the inherent imbalance in the distribution of SM observations, as samples at high SM levels are naturally scarce across both time and space. In addition, satellite observations under wet conditions tend to be less reliable because of saturation effects, cloud contamination, and decreased sensitivity in albedo, all of which affect the quality of the input features. Despite these challenges, the AtLSTM model still achieves much better performance than XGBoost in this range, as demonstrated by significantly improved R2 and RMSE values. In general, the model performs well across the majority of the SM range, with only slight differences in overall accuracy between the spatially independent training and test sets, indicating relatively robust generalization capability.

In response, the following sentences have been added to the revised manuscript:

"These samples were then randomly divided into training, validation, and test datasets in the ratio of 7:2:1 based on their locations, ensuring spatial independence with distances between any two samples exceeding 25 km, thereby minimizing the influence of spatial autocorrelation."

"In particular, the reduced model accuracy observed in the high SM range is likely due to the inherent imbalance in the numerical distribution of SM samples and increased uncertainty in the accuracy of input features under wet surface conditions."

3. Figure 2 in the Methods section is unnecessarily technical and lacks clear justification.

A concise textual explanation that hyperparameter tuning was performed would be sufficient. Moreover, both Figures 1 and 2 fail to clearly illustrate how the spatiotemporal training was implemented, especially considering that LSTM and related models are specifically designed to capture temporal dependencies. The authors should provide a more explicit and structured explanation of how time-series characteristics were incorporated into the model training, without ignoring the spatial autocorrelation issue raised above.

Thank you for your valuable comments. In response, we have added a new schematic diagram (Fig. 2) to explicitly illustrate how spatiotemporal sampling was conducted from the input dataset to construct time-series input samples that are evenly distributed across the globe. Specifically, spatial sampling was performed at an interval of 5 pixels, ensuring that the minimum distance between any two samples exceeds 25 km. As a result, the training and test samples are spatially independent, and the potential influence of spatial autocorrelation is expected to be minimal.

---

## Author Response (AR2)

**Topic editor comment**

The manuscript presents a promising approach with thoughtful revisions that have addressed many of the initial concerns. However, a key issue remains unresolved and was raised by one of the reviewers regarding the justification for using ERA5-Land predictors. To ensure the robustness of the study's conclusions, an additional experiment excluding ERA5-Land is necessary to clarify whether performance improvements are driven by the modeling approach or the input data quality. A major revision is therefore warranted to address this methodological uncertainty.

Thank you for your time and helpful feedback. We have conducted the ablation experiment by excluding the ERA5-Land SM as suggested by the reviewer. Please refer to our detailed response to the reviewer for the experimental results and discussion.

**Report #2**

Thank you for the authors' thoughtful responses to my previous comments. I agree with most of the clarifications and revisions made.

However, the justification for using ERA5-Land as a predictor remains insufficient. It is still unclear whether the improved model performance is primarily due to the advanced methods employed or the quality of the ERA5-Land dataset itself. To disentangle these effects, I strongly suggest that the authors conduct an additional experiment testing model performance without the ERA5-Land predictors. This would help assess the degree to which ERA5-Land contributes to the overall results and improve the robustness of the conclusions.

We sincerely appreciate your valuable suggestion. In response, we conducted an additional experiment by excluding the ERA5-Land SM from the input features and re-evaluated the performance of the AtLSTM model on the test set. The results show that the R² decreased from 0.987 to 0.954, and the RMSE increased from 0.011 $m^3$ $m^{-3}$ to 0.020 $m^3$ $m^{-3}$. Similarly, when the GLASS-AVHRR albedo and LST were removed from the inputs, the R² dropped to 0.968 and the RMSE increased to 0.018 $m^3$ $m^{-3}$. These results demonstrate that both ERA5-Land SM and GLASS-AVHRR albedo and LST are critical to the performance of our long-term SM estimation model. By integrating multi-source datasets and leveraging their complementary strengths, the model achieves substantially improved accuracy on the test set. As data-driven approaches, the performance of ML and DL models is highly dependent on the quality of the input datasets. Nevertheless, the comparative analysis conducted in this study indicates that utilizing temporal information and optimizing model architectures also

play an important role in further enhancing the accuracy of the SM estimation model.

Additionally, we have added the following sentences to the revised manuscript: "To further investigate the importance of multi-source datasets for the performance of the AtLSTM model, we conducted ablation experiments by individually removing the ERA5-Land SM and the GLASS-AVHRR albedo and LST products from the input datasets. The results show that the AtLSTM model's accuracy on the test set decreased significantly, with R² dropping to 0.954 and 0.968, and RMSE increasing to 0.020 m$^3$ m$^{-3}$ and 0.018 m$^3$ m$^{-3}$, respectively. These results demonstrate that by integrating multi-source datasets and leveraging their complementary strengths, the AtLSTM model cam achieve substantially improved accuracy in long-term SM estimation." (P16, L400-406)